# The physiological landscape and specificity of antibody repertoires are consolidated by multiple immunizations

**Lucia Csepregi[1], Kenneth Hoehn[2], Daniel Neumeier[1], Joseph M Taft[1], Simon Friedensohn[1,3], Cédric R Weber[1,3], Arkadij Kummer[1], Fabian Sesterhenn[4], Bruno E Correia[4,5], Sai T Reddy[1]***

[1]Department of Biosystems Science and Engineering, ETH Zürich, Basel, Switzerland; [2]Department of Pathology, Yale University School of Medicine, New Haven, United States; [3]Alloy Therapeutics AG, Basel, Switzerland; [4]Institute of Bioengineering, École Polytechnique Fédérale de Lausanne, Lausanne, Switzerland; [5]SIB Swiss Institute of Bioinformatics, Lausanne, Switzerland

**\*For correspondence:**
sai.reddy@bsse.ethz.ch

**Abstract** Diverse antibody repertoires spanning multiple lymphoid organs (i.e., bone marrow, spleen, lymph nodes) form the foundation of protective humoral immunity. Changes in their composition across lymphoid organs are a consequence of B-cell selection and migration events leading to a highly dynamic and unique physiological landscape of antibody repertoires upon antigenic challenge (e.g., vaccination). However, to what extent B cells encoding identical or similar antibody sequences (clones) are distributed across multiple lymphoid organs and how this is shaped by the strength of a humoral response remains largely unexplored. Here, we performed an in-depth systems analysis of antibody repertoires across multiple distinct lymphoid organs of immunized mice and discovered that organ-specific antibody repertoire features (i.e., germline V-gene usage and clonal expansion profiles) equilibrated upon a strong humoral response (multiple immunizations and high serum titers). This resulted in a surprisingly high degree of repertoire consolidation, characterized by highly connected and overlapping B-cell clones across multiple lymphoid organs. Finally, we revealed distinct physiological axes indicating clonal migrations and showed that antibody repertoire consolidation directly correlated with antigen specificity. Our study uncovered how a strong humoral response resulted in a more uniform but redundant physiological landscape of antibody repertoires, indicating that increases in antibody serum titers were a result of synergistic contributions from antigen-specific B-cell clones distributed across multiple lymphoid organs. Our findings provide valuable insights for the assessment and design of vaccine strategies.

## Editor's evaluation

This study provides an important systems analysis of antibody repertoires across multiple lymphoid organs, demonstrating significant clonal overlap following repeated immunizations. The findings show that strong humoral responses lead to a high degree of repertoire consolidation, correlating with antigen specificity and B-cell migration between organs. The evidence is convincing, with deep sequencing and network analyses strongly supporting the conclusions.

## Introduction

B cells exert their protective functions via their B-cell receptors (BCRs) (and secreted version: antibodies), which are capable of recognizing a plethora of antigenic epitopes and structures.

Hematopoietic precursors differentiate into immature B cells in the bone marrow, which then migrate and mature to naive B cells in secondary lymphoid organs such as the spleen, lymph nodes, and mucosa-associated lymphoid tissue, which are distributed throughout the body and are interconnected through the circulatory and lymphatic systems. Naive B cells, which are continuously circulating and screening for antigens, can become activated upon an encounter with a cognate antigen (BCR binding to antigen), which results in differentiation into memory B cells and plasma cell subsets that eventually migrate to supportive niches for their long-term survival (*Ellyard et al., 2004*; *Wilmore and Allman, 2017*). Thus, the composition of antigen-specific B cells throughout the body is highly dynamic on a molecular, cellular, and spatio-temporal level, resulting in a highly diverse and unique physiological landscape of antibody repertoires (*Batista and Harwood, 2009*).

To assess antigen-induced B-cell responses, measurements of serum antibody titers can be performed, which are mainly reflective of circulating antibodies (immunoglobulins [Ig]) secreted from long-lived plasma cells (LLPCs) that reside in the bone marrow, spleen, and lymph nodes (*Ellyard et al., 2004*; *Manz et al., 2005*; *Slifka et al., 1998*). However, it is still unknown how the strength of humoral immunity, which is correlated with serum antibody titers (e.g., IgG), is reflected in the physiological composition of antibody repertoires present across multiple lymphoid organs.

Thus far, only a few systematic studies have profiled antibody repertoires across multiple organs within the same individuals. In humans, due to limited organ accessibility, most studies performed antibody repertoire analysis on B cells derived from peripheral blood, with a few notable exceptions analyzing more than two organs (*Briney et al., 2014*; *Domínguez Conde et al., 2022*; *Mandric et al., 2020*; *Meng et al., 2017*; *Jones et al., 2022*; *Yang et al., 2021*). For example, Meng et al. constructed an atlas of B-cell clonal distribution uncovering distinct evolution of clonal lineages in human tissues such as blood-rich compartments versus the gastrointestinal tract (*Meng et al., 2017*). Yang et al. reported shared antibody repertoires across human tissues and inferred antigen specificity based on similarity to previously reported pathogen-specific antibody sequences (*Yang et al., 2021*). In the context of mice, Mathew et al. analyzed antigen-specific B cells in the draining lymph nodes, spleen, and lungs after influenza infection, which revealed distinct organ-specific antibody repertoire features (*Mathew et al., 2021*). However, similar to other studies across tissues in mice, they performed small-scale sequencing on specific B-cell subsets (*Mesin et al., 2020*; *Riedel et al., 2020*; *Tas et al., 2016*). Therefore, it remains an open question how the degree of clonal overlap across physiological compartments relates on repertoire scale and whether each lymphoid organ represents an exclusive or redundant (class-switched IgG) antibody repertoire upon antigenic challenge.

Here, we map the physiological landscape of antibody repertoires in mice following immunization. We used a systems-based approach that included deep and single-cell sequencing, bioinformatic and statistical analysis, and high-throughput antibody specificity screening to comprehensively profile antibody repertoires from six distinct lymphoid organs. Our analysis uncovered that strong humoral responses (high antibody titers due to repeated immunizations) led to the physiological consolidation of antibody repertoires (a high fraction of clones shared across multiple lymphoid organs). Additionally, clustering and phylogenetic analysis revealed connections between organs such as the spleen and bone marrow at the repertoire level, suggesting B-cell migration events that contribute to substantial repertoire consolidation. Notably, we observed a correlation between the clonal overlap across organs and antigen-binding specificity. These findings, which demonstrate that strong humoral immunity corresponds to a more uniform and homogenous physiological antibody repertoire, may provide important insights for the assessment and design of vaccine strategies.

## Results
### Systems analysis of the physiological landscape of antibody repertoires

To profile the physiological landscape of antibody repertoires, we performed a comprehensive systems analysis that included murine models of immunization, antibody repertoire sequencing, bioinformatic repertoire analysis, and high-throughput antibody specificity screening (*Figures 1a* and 5a and g). The study was designed around two cohorts of mice that either received a single immunization (cohort-1x, consisting of mice 1x-A, 1x-B, and 1x-C) or multiple booster immunizations (cohort-3x, mice 3x-D, 3x-E, and 3x-F). Mice were subcutaneously injected at the left lateral flank with viral protein antigen (pre-fusion glycoprotein of respiratory syncytial virus [RSV-F]) in poly(I:C) adjuvant, with cohort-3x

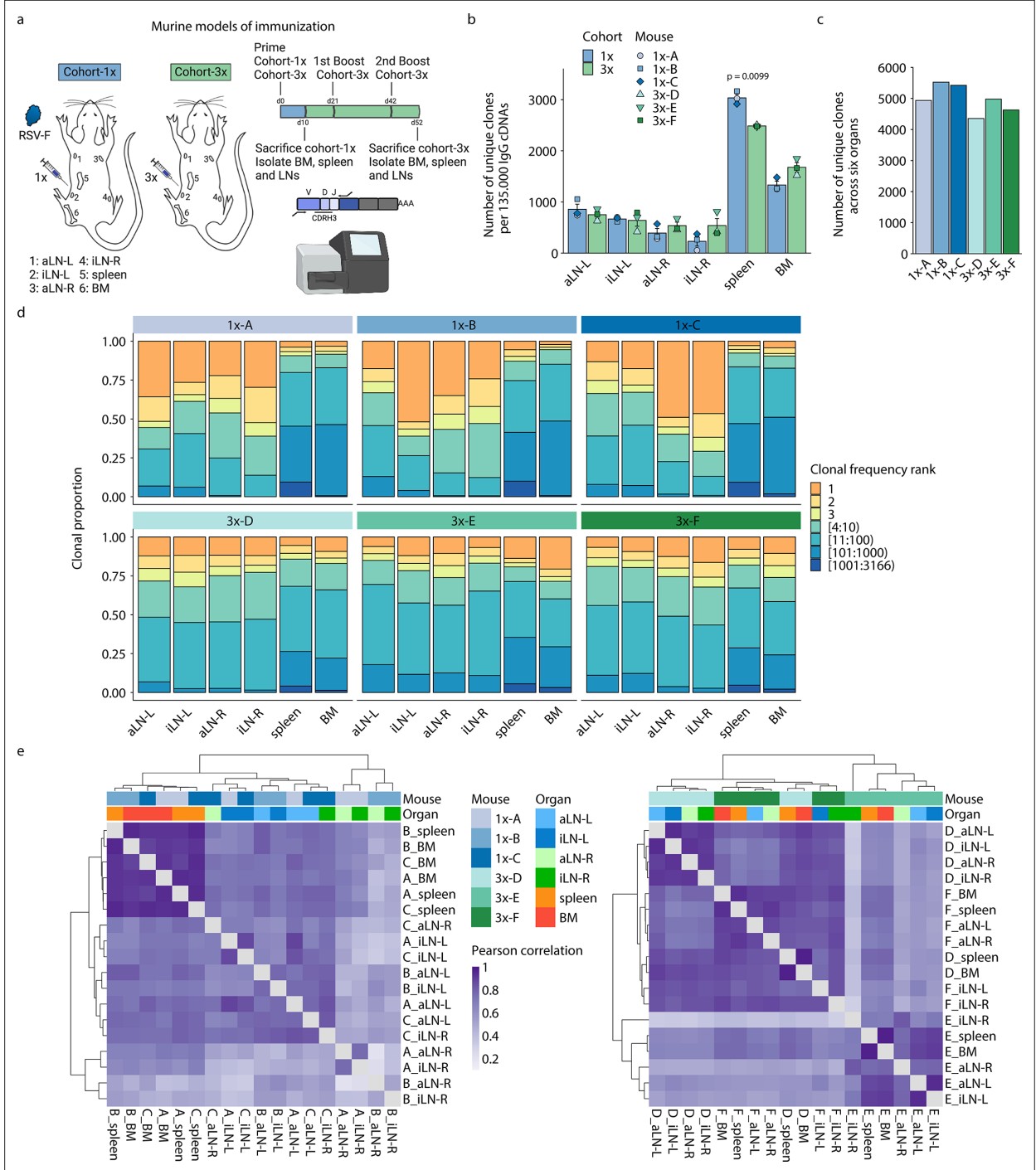

**Figure 1.** Profiling antibody repertoire diversity and clonal expansion across multiple lymphoid organs. (**a**) Overview of mouse immunizations with RSV-F antigen: Mice (n = 3 per cohort) were subjected to single (cohort-1x) or multiple (cohort-3x) immunizations (subcutaneous injections into the left lateral flank), followed by deep sequencing of antibody repertoires (VH IgG) derived from six different lymphoid organs (BM, spleen, aLN-L, -R, and iLN-L, -R). (**b**) Shown is the average clonal diversity (mean ± SEM, n = 3) based on the total number of unique clones (clones were defined as antibody sequences possessing identical germline V- and J-genes and 100% CDRH3 amino acid [a.a.] identity and length) within each organ per mouse (point) and cohort (bar). p-Values represent the results from unpaired *t*-test, adjusted for multiple testing using the Benjamini–Hochberg method. The comparison was made across cohorts for the same organs, and only p-values <0.05 are depicted. (**c**) Bar plots represent the clonal diversity by quantifying the total numbers of unique clones across all organs within each mouse. (**d**) Clonal expansion profiles with each bar representing an organ and each colored section displaying frequency-ranked clones or clonal fractions and the corresponding percentage they occupy within each organ repertoire. (**e**) Hierarchical clustering of Pearson correlation coefficients of germline V-gene usage counts between organs. Cohort-1x and cohort-3x were separately

*Figure 1 continued on next page*

*Figure 1 continued*

clustered (left and right). Each tile represents the pairwise Pearson correlation coefficient of the germline V-gene repertoires of two organs. BM: bone marrow; aLN-L, -R: left and right axillary lymph nodes; iLN-L, -R: left and right inguinal lymph nodes. Panel (**a**) created with BioRender.com.

The online version of this article includes the following figure supplement(s) for figure 1:

**Figure supplement 1.** Serum titer measurements, evaluation of clonotyping, quantification of clonal diversities, and subsampling analysis.

**Figure supplement 2.** Average Hill-based evenness profiles, Shannon indices, IgM expansion profiles, and Pearson correlation of germline V-gene usage.

mice receiving three injections at 21-day intervals (*Figure 1a*), as it is well established that multiple booster immunization schemes induce strong humoral immune responses (*Manz et al., 2005*; *Manz et al., 1997*). For both cohorts, mice were sacrificed 10 days after the final immunization, followed by isolation of six lymphoid organs: bone marrow, spleen, and left and right axillary and inguinal lymph nodes. Antigen-binding titers from serum confirmed that cohort-3x mice indeed had a strong antibody response (average endpoint titer >1/1 × 10$^6$) relative to the weaker response of cohort-1x mice (average endpoint titer ~1 /3 × 10$^4$) (*Figure 1—figure supplement 1a*).

Deep sequencing of antibody repertoires has emerged as a powerful tool for assessing important immunological parameters such as clonal diversity, expansion, and evolution (somatic hypermutation) and thus offers a quantitative approach to profile antibody repertoires from specific tissues (*Bashford-Rogers et al., 2019*; *Briney et al., 2019*; *Greiff et al., 2017a*; *Soto et al., 2019*). In order to reveal the physiological landscape of antibody repertoires following immunization, we performed targeted deep sequencing of six lymphoid organs in all three mice of both cohorts (total of 36 organs). Variable heavy (VH) chain IgG sequencing libraries were prepared according to our previously established protocol that incorporates unique molecular identifiers for error and bias correction (*Khan et al., 2016*). Focusing on the IgG-class-switched repertoire allowed for a pooled analysis of mainly plasmablasts and plasma cells due to their higher rate of antibody gene transcription. Consequently, our study characterizes the distribution and repertoire features of mainly expanded B-cell clones. Deep sequencing (Illumina MiSeq, paired-end 2 × 300 bp) and preprocessing resulted in an average of 6 × 10$^5$ reads per organ and up to 4 × 10$^5$ quality processed and merged reads per organ were used as input for the error-correction pipeline (*Supplementary file 1*).

## Antibody repertoire analysis across lymphoid organs

First, we aimed to characterize common antibody repertoire features (i.e., clonal diversity, expansion, and germline V-gene usage) in each lymphoid organ for both immunization cohorts. For this, we defined B-cell clones as antibody sequences possessing identical germline V- and J-genes and 100% CDRH3 amino acid (a.a.) identity and length. For investigation of groups of clonally related B-cell variants (here referred to as clonotypes), antibody sequences were clustered together in the same manner with the exception of using 90% CDRH3 a.a. identity (similar ratios between clones and clonotypes were observed when using 80% or 85% CDRH3 a.a. identity, see *Figure 1—figure supplement 1b*).

Quantification of the total number of unique clones within each organ repertoire showed that lymph nodes displayed less diverse antibody repertoires (55–1055 unique clones) than bone marrow (1252–1846 unique clones) and spleen (2475–3166 unique clones) across all mice (*Figure 1b*; for diversity of unique clonotypes, see *Supplementary file 2* and *Figure 1—figure supplement 1c*). Although lymph nodes are comparatively small, the lower diversity of IgG+ clones observed may not be a consequence of their size, as the clonal diversity of IgM+ clones in lymph nodes was not reduced compared to spleen and bone marrow in untreated mice (*Figure 1—figure supplement 1e*). In cohort-1x mice, the left lymph nodes (side of immunization) displayed slightly higher diversities compared to the right lymph nodes. This difference was not visible in cohort-3x mice, which exhibited comparable diversities in lymph nodes from both sides. Moreover, while the bone marrow repertoires of cohort-3x mice displayed an increase in clonal (but not clonotype) diversity, the spleen repertoires displayed a decrease in both clonal and clonotype diversity. Pooling all clones/clonotypes derived from all six lymphoid organs and quantifying all unique clones/clonotypes within each mouse revealed that cohort-3x mice had a reduced antibody repertoire diversity per mouse (*Figure 1c*, *Figure 1—figure supplement 1d*). To account for differences in sequencing input across each organ, subsampling analysis was performed with varying sequence input into the error-correction pipeline, which

confirmed the observed differences in diversity profiles across organs (*Figure 1—figure supplement 1f and g*).

Next, we assessed the state of clonal expansion in all organ repertoires using Hill-based evenness profiles (*Figure 1—figure supplement 2a and b*) – a diversity measure derived from mathematical ecology and commonly used in immune repertoire analysis (*Greiff et al., 2017b*; *Greiff et al., 2015*; *Hill, 1973*). Hill-based evenness profiles indicate the distribution of clonal frequencies within each repertoire and enable comparison of the extent of clonal expansion across multiple repertoires (evenness values range from 0 and 1, with values closer to 1 representing a uniform clonal frequency distribution, while values closer to 0 indicate the prevalence of one or a few highly expanded clones; for more details, see 'Materials and methods'). The analysis revealed the highest degree of clonal expansion in lymph nodes (mean Shannon evenness $^{\alpha=1}E$ of 0.06) compared to spleen (mean Shannon evenness $^{\alpha=1}E$ of 0.17), and bone marrow (Shannon evenness $^{\alpha=1}E$ of 0.33) in cohort-1x mice. By contrast, a high degree of clonal expansion was uniformly found across all organs in cohort-3x mice (with mean Shannon evenness $^{\alpha=1}E$ values of 0.12, 0.10, and 0.10 in lymph nodes, spleen, and bone marrow, respectively).

To further dissect what fraction of clones contributed to the observed clonally expanded (polarized) repertoires, we visualized the frequencies for specific proportions of clones present in each organ (*Figure 1d*). Notably, both left and right lymph node repertoires of cohort-1x mice were dominated by a few highly expanded clones (indicated in orange and yellow). For instance, the three most frequent clones comprised 33% to 71% of the whole repertoire in all three mice. This stands in contrast to the spleen and bone marrow, in which the three most frequent clones occupied only up to 13% and 10%, respectively. Strikingly, compared to cohort-1x, expansion profiles in cohort-3x were more uniform across all organs, with the three most expanded clones displaying frequencies ranging from 15% to 32% in lymph nodes and up to 19% and 29% in spleen and bone marrow, respectively. Lymphoid organs of untreated mice did not show such degree of expansion within IgM+ clones in lymph nodes and bone marrow, with the three most frequent clones displaying frequencies ranging from 1.2% to 6.8% in lymph nodes and 3.9% to 7.9% in bone marrow (*Figure 1—figure supplement 2c*). The spleens of untreated mice displayed reduced clonal diversity (*Figure 1—figure supplement 1e*), while exhibiting increased clonal expansion compared to lymph nodes and bone marrow, with the three most frequent clones showing frequencies of 9.1%, 13.9%, and 11.3% in mouse 1, 2, and 3, respectively (*Figure 1—figure supplement 2c*).

Next, quantification of germline V-gene usage of cohort-1x and cohort-3x mice revealed that clones from spleen and bone marrow contained more unique V-genes (99–115) compared to clones from lymph nodes (31–92), in line with the observed reduction of clonal diversity in lymph nodes of all mice. Hierarchical clustering of Pearson correlation coefficients for each mouse showed that spleen and bone marrow displayed the highest correlation in germline V-gene usage across both cohorts (*Figure 1—figure supplement 2d*). In two out of three mice per cohort, the left lymph nodes as well as right lymph nodes clustered together, respectively, possibly reflecting a physiological axis upon immunization. Moreover, we observed cohort-specific differences in germline V-gene usage (*Figure 1e*). In cohort-1x mice, the same organs of different mice clustered together, with the spleen and bone marrow displaying the highest similarity. By contrast, in cohort-3x mice, organs within the same mouse clustered together, reflecting a physiological consolidation of germline V-gene repertoires.

## Strong humoral responses result in highly connected and overlapping clones

In addition to clonal expansion, antigen immunization drives affinity maturation by somatic hypermutation, which results in production of clonal variants. To understand the network of clonal variants, we constructed sequence-similarity networks to elucidate clonal relationships (*Bashford-Rogers et al., 2013*; *Miho et al., 2019*). Networks were generated where clones (nodes) within a clonotype were connected (edges) if they differed by one a.a. in CDRH3 (*Figure 2*, legend). When assessing clonal connectivity by quantifying the number of edges per node, cohort-3x mice displayed an increased fraction of clones with higher connectivity compared to cohort-1x mice (*Figure 2—figure supplement 1a*). Network visualization of the five most diverse clonotypes per organ revealed that cohort-3x mice contained a substantially higher clonal connectivity and diversity across all organs, in particular for the spleen and bone marrow (*Figure 2*, *Figure 2—figure supplement 1b and c*). These observations

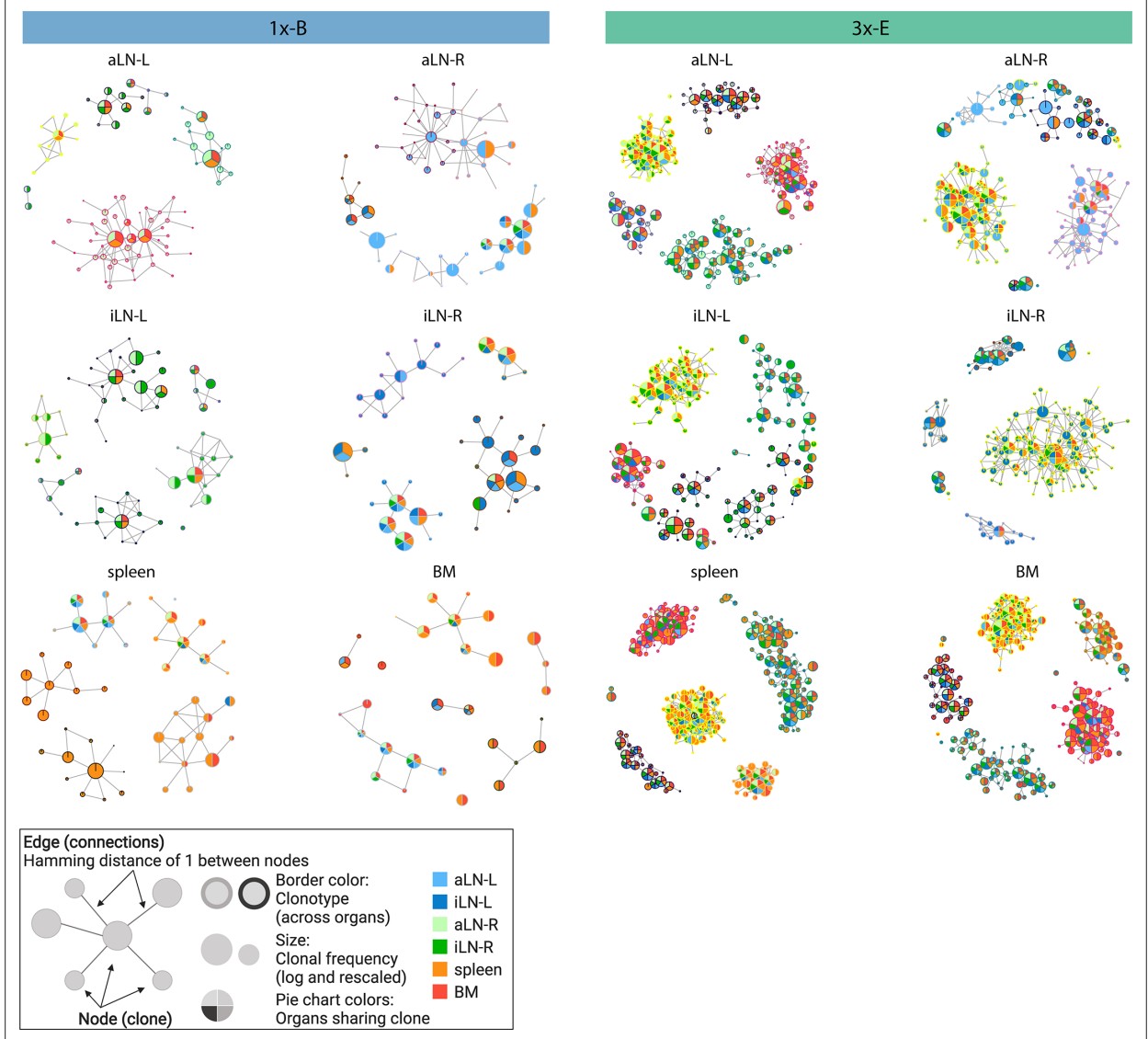

**Figure 2.** Sequence-similarity network analysis across lymphoid organs. Sequence-similarity networks of the five most diverse clonotypes from a representative mouse from the single (1x-B) or multiple (3x-E) immunization cohorts. Within a network, each node represents a unique clone that is connected by edges to clones with a Hamming distance of 1 between CDRH3 a.a. Clones within each clonotype display the same border color throughout all organs in each mouse. The size of each clone represents the clonal frequency (in log) within the corresponding organ. Pie charts within each node depict organs sharing the corresponding clone (see legend). See *Figure 2—figure supplement 1c* for sequence similarity networks of the other cohort-1x and cohort-3x mice. BM: bone marrow; aLN-L, -R: left and right axillary lymph nodes; iLN-L, -R: left and right inguinal lymph nodes. Legend created with BioRender.com.

The online version of this article includes the following figure supplement(s) for figure 2:

**Figure supplement 1.** Quantification of clonal connectivity, clonal diversity and network-based analysis of the five most diverse clones.

**Figure supplement 2.** Quantification of somatic hypermutation (SHM).

were in line with higher somatic hypermutation counts across all organs in cohort-3x mice (*Figure 2—figure supplement 2a and b*). Notably, networks of the five most diverse clonotypes of both cohorts consisted of low- and high-frequency clones, with frequencies ranging from $5 \times 10^{-4}$% and $6 \times 10^{-4}$% to 22% and 21% within the respective organs in cohort-1x and cohort-3x mice, respectively. Moreover, within cohort-3x mice, a substantial fraction of the five most diverse clonotypes were shared across all six organs (*Supplementary file 3*), and the majority of clones were present in all organs (*Figure 2*, *Figure 2—figure supplement 1c*).

To gain deeper insights into global repertoire similarities across lymphoid organs, we performed extensive clonal overlap analyses. Cohort-1x mice shared only 1–3 clones, while cohort-3x mice shared 49–71 clones across all six organs (*Figure 3a*, *Figure 3—figure supplement 1a*). Next, by quantifying shared clones across organs within each mouse (requiring identical clones in at least two organs), we observed that in cohort-1x mice the majority of clones were found in a single organ (organ-exclusive). In contrast, cohort-3x mice displayed a substantial fraction of clones shared across lymphoid organs, particularly in lymph nodes (indicated in gray by pie charts, *Figure 3b*). To reveal the extent of shared clones between organ pairs, we calculated the pairwise Jaccard index (by dividing the number of unique clones shared between two organ repertoires by the total number of unique clones present in both repertoires; indicated by the thickness of the connective lines; *Figure 3b*, *Figure 3—figure supplement 1b*) as well as pairwise cosine indices (*Figure 3c*, *Figure 3—figure supplement 2*). In contrast to the Jaccard index, which is only based on clonal identity, the cosine similarity takes into account clonal overlap as well as clonal frequency (repertoires with cosine indices near to 1 indicate high pairwise repertoire similarity based on both clonal overlap and clonal frequency). Both similarity measurements revealed that cohort-3x mice showed a greater degree of repertoire similarity across all organs. This is consistent with the observation of higher similarities in germline V-gene usage across organs in cohort-3x mice (*Figure 1e*). Physiological segregation of clones was observed with the left lymph nodes, right lymph nodes, as well as the spleen and bone marrow displaying increased repertoire similarities (based on germline V-gene usage [*Figure 1—figure supplement 2d*] and clonal overlap [*Figure 3b and c*]) compared to other organs across both cohorts. Of note, mouse 1x-C showed high pairwise cosine similarity between the right inguinal and axillary lymph nodes because both organs contained an identical clone (CDRH3: CARSLYGAFDYW) representing 47% and 49% of the repertoire, respectively (*Figure 3c*). Focusing on the 10 most frequent clones for each organ showed that shared clones also displayed similar frequencies across organs in cohort-3x and that increased overlap was also detected in low frequency clone fractions compared to cohort-1x (*Figure 3—figure supplement 3*).

## Phylogenetic analysis reveals physiological axes of clonal evolution

We next aimed to understand the evolution of B cells across all six lymphoid organs during weak and strong humoral responses. To trace the evolution of clones across multiple organs, we inferred lineage trees for each clonotype using *IgPhyML* (*Hoehn et al., 2019*). Lineage trees containing clonal sequences from multiple organs may represent clonotypes that have migrated among organs, and the distribution of organs along the tips of B-cell lineage trees can provide important information about the nature of these cellular migrations. Visual inspection of the largest clonotypes in each mouse showed that in cohort-1x mice, clonotypes were typically restricted to either left or right lymph nodes, with some sequences found in the spleen and bone marrow (*Figure 4a*, *Figure 4—figure supplement 1a*). By contrast, large clonotypes in cohort-3x mice were more evenly dispersed among all six organs. These results are consistent with prior results showing a high degree of clonal overlap among left lymph nodes, right lymph nodes, and spleen/bone marrow in cohort-1x mice, with more widespread clones across all organs in cohort-3x mice (*Figure 3*, *Figure 3—figure supplement 3*).

To further investigate the spread of clonotypes among lymphoid organs, we used the recently developed switch proportion (SP) test, which compares the proportion of each type of organ transition along observed trees to those obtained from trees with randomly assigned organ locations (*Hoehn et al., 2022*). A greater than expected proportion of transitions in a particular direction along lineage trees can be indicative of cellular migration patterns. We found that lineage trees in all cohort-1x mice had a significantly greater proportion of transitions from spleen to bone marrow than expected with random switching (*Figure 4b*). In contrast, we found that lineage trees of all cohort-3x mice had a significantly higher proportion of transitions both from spleen to bone marrow, and from bone marrow to spleen. This indicates a close association between these organs in cohort-3x trees that is not biased in a particular direction. Repeating these analyses using only clonal sequences from spleen and bone marrow confirmed these patterns: cohort-1x mice had significantly more transitions from spleen to bone marrow, while cohort-3x mice did not (*Figure 4—figure supplement 1b*). To detect associations between lymphoid organs within lineage trees, we repeated the SP test but permuted organ labels among trees within each mouse and quantified transitions in either direction between tissues. Consistent with prior results, we found a significantly greater proportion of switches between spleen

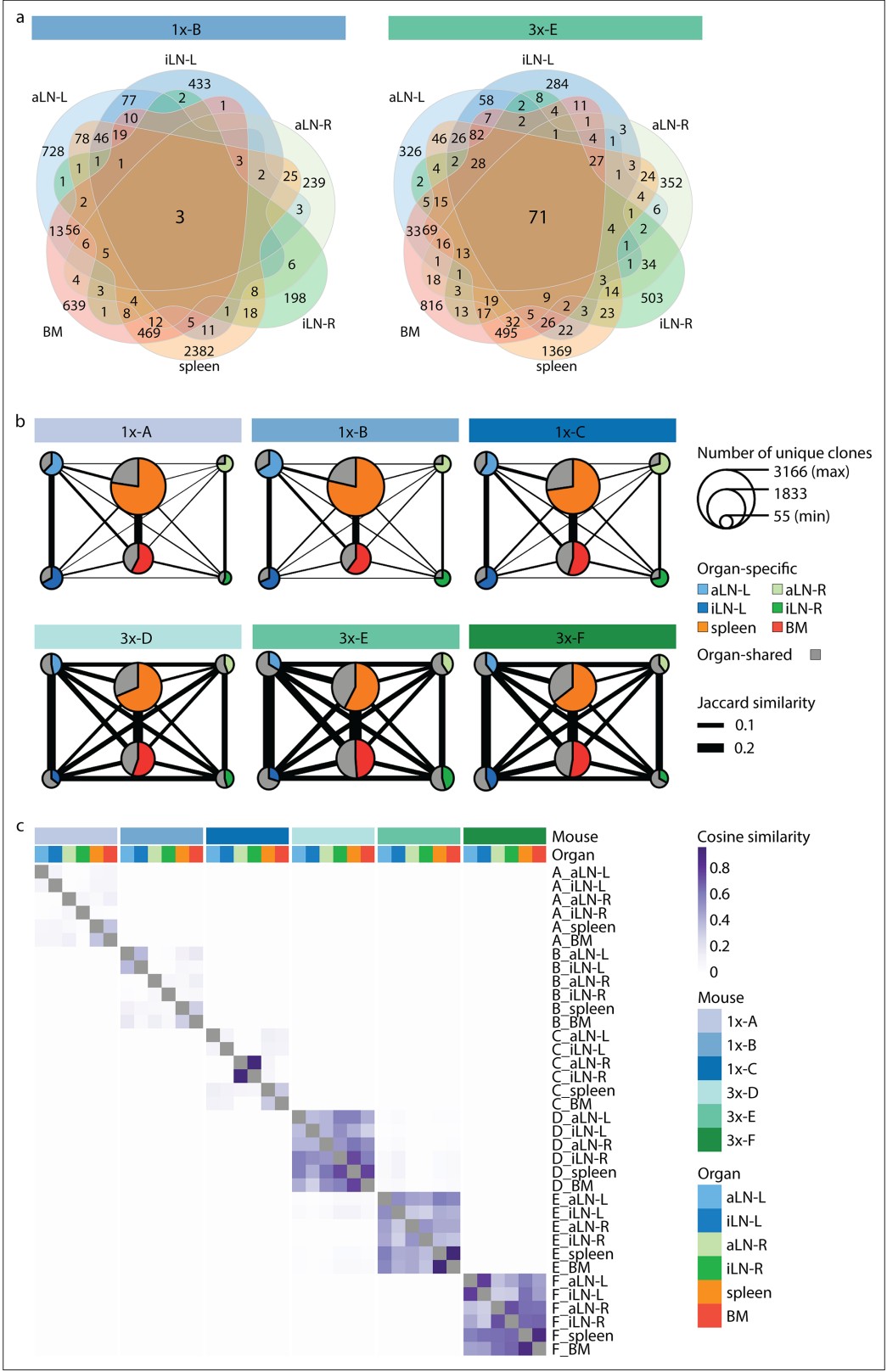

**Figure 3.** Strong humoral responses result in increased antibody repertoire similarities across multiple lymphoid organs. (**a**) Venn diagrams depicting numbers of shared unique clones from a representative mouse from the single (1x-B) or multiple (3x-E) immunization cohorts. See *Figure 3—figure supplement 1a* for Venn diagrams of the other cohort-1x and cohort-3x mice. (**b**) Repertoire similarity networks are based on clonal overlap across all

*Figure 3 continued on next page*

*Figure 3 continued*

six lymphoid organs per mouse. Each node represents an organ, with the size of the node being proportional to the number of unique clones within the organ. Width of edges between the nodes depict the pairwise repertoire similarity based on the Jaccard index (see 'Materials and methods'). The pie chart within each node represents the proportion of organ-specific and organ-shared clones for each organ. (**c**) Heatmap representing pairwise cosine similarity indices based on clonal overlap including clonal frequency information across all samples. BM: bone marrow; aLN-L, -R: left and right axillary lymph nodes; iLN-L, -R: left and right inguinal lymph nodes.

The online version of this article includes the following figure supplement(s) for figure 3:

**Figure supplement 1.** Repertoire similarity based on clonal overlap.

**Figure supplement 2.** Repertoire similarity based on pairwise cosine similarities.

**Figure supplement 3.** Tracking of diversity ranked clones across organs.

and bone marrow in all mice (*Figure 4—figure supplement 1c*). We also observed a distinct left/right axis among lymph nodes, with five of six mice having a significantly greater proportion of transitions among left axillary and inguinal lymph nodes (*Figure 4—figure supplement 1c*). In contrast, only two of six mice (both in cohort-1x) had a significantly greater proportion of switches between right-side lymph nodes. Because all mice were immunized on the left side, the greater connectivity among left-side lymph nodes may be due to stronger localized immune responses on the left side. Overall, these phylogenetic analyses provide further evidence that B-cell clonotypes are more segmented along left, right, and spleen/bone marrow axes upon a single immunization (at the left lateral flank), but are more widely dispersed following multiple immunizations.

## Physiological consolidation of antibody repertoires correlates with antigen specificity

After uncovering that a strong humoral response in cohort-3x mice induced physiological consolidation of antibody repertoires, we next aimed to determine how this finding translates to antigen specificity. Interrogating antigen specificity on a repertoire scale requires experimental screening platforms capable of linking antibody genotype and phenotype (antigen-binding) (*Adler et al., 2017*; *Asensio et al., 2019*; *Parola et al., 2019*; *Venet et al., 2013*; *Wang et al., 2016*; *Wang et al., 2015*). Therefore, we utilized yeast surface display (*Feldhaus et al., 2003*), fluorescence-activated cell sorting (FACS), and deep sequencing to identify antigen-specific clones present in antibody repertoires (*Figure 5a*). We generated three single-chain variable fragment (scFv) libraries from each of the cohort-3x mice by combinatorial (random) pairing of VH and variable light chain (VL) genes derived from bone marrow (*Figure 5a*, *Figure 5—figure supplement 1a and b*). As there was a high degree of clonal overlap across all organs in cohort-3x mice, we decided that bone marrow would be a representative organ while also providing ample material for scFv library generation due to the presence of a high fraction of plasma cells expressing large amounts of IgG RNA (*Reddy et al., 2010*; *Shi et al., 2015*). scFv yeast display libraries were enriched for RSV-F antigen-binding by FACS over multiple rounds (*Figure 5—figure supplement 1c*). By using a stringent FACS gating strategy, we ensured maximal purity at some expense of diversity, presumably by filtering out low-affinity clones. The final sorted population of yeast cells with specificity to antigen were subjected to deep sequencing of VH genes and the resulting CDRH3s were matched with the repertoire data of all organs within each corresponding mouse to assign RSV-F specificity to antibody clones (now referred to as 'binding clones').

This workflow resulted in the detection of 120 unique binding clones from the three mice in cohort-3x, with the following distribution of binding clones (clonotypes) in each mouse: mouse 3x-D = 42 (13), 3x-E = 59 (23), and 3x-F = 22 (15). Importantly, 88–95% of these binding clones were shared between at least two organs out of which 24–41% were overlapping across all six lymphoid organs within the corresponding mouse (*Figure 5b*). Since binding clones were experimentally derived from bone marrow RNA, the fact that the vast majority of clones (74–86%) were also observed in spleen confirmed the high degree of repertoire similarity in these two organs; additionally, a substantial fraction of binding clones was also observed in lymph nodes (51–68%) (*Figure 5c*).

Strikingly, the binding clones that were shared across all six organs occupied on average 24%, 24%, and 22% of the corresponding organ repertoire in each mouse, respectively (*Figure 5d*). When quantifying all clonal variants of clonotypes that were associated with the binding clones, left lymph

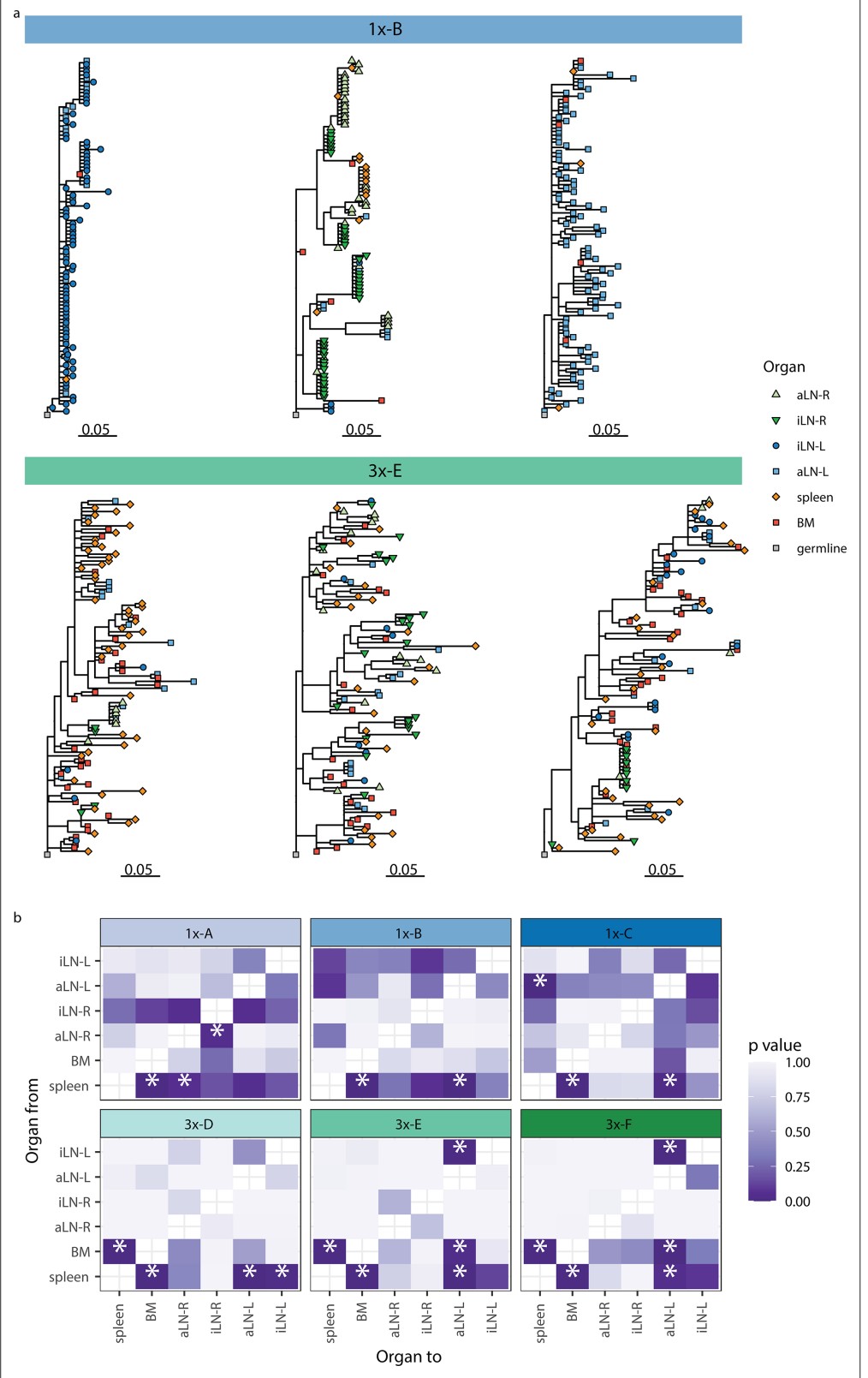

**Figure 4.** Phylogenetic analysis reveals physiological axes, diversification, and sharing of clonotypes across multiple lymphoid organs. (**a**) Phylogenetic trees of the three most diverse clonotypes from a representative mouse from the single (1x-B) or multiple (3x-E) immunization cohorts. Color of nodes indicates the lymphoid organ origin of each tip with gray nodes depicting the clonotype's unmutated germline sequence. Scale bar shows somatic

*Figure 4 continued on next page*

*Figure 4 continued*

hypermutations per codon site. See *Figure 4—figure supplement 1a* for phylogenetic trees of the other cohort-1x and cohort-3x mice. (**b**) Heatmaps displaying p-values derived from switch proportion (SP) tests, which quantify enrichment of transitions from one organ to another within lineage trees. BM: bone marrow; aLN-L, -R: left and right axillary lymph nodes; iLN-L, -R: left and right inguinal lymph nodes. Asterisks indicate an SP test p<0.05.

The online version of this article includes the following figure supplement(s) for figure 4:

**Figure supplement 1.** Phylogenetic analysis.

nodes displayed comparable or even higher numbers of unique clonal variants than right lymph nodes, whereas the spleen and bone marrow showed consistently the highest numbers of clonal variants (*Figure 5—figure supplement 2a*). Of note, we detected a shared (public) binding clonotype present in all six lymphoid organs of all three mice, thus hinting toward physiological consolidation and sequence convergence of antibody repertoires (*Figure 5—figure supplement 2b*).

By also accounting for clonal expansion (clonal frequencies), we observed that the extent of organ sharing in binding clones positively correlated with higher clonal frequencies; that is, clones shared across more organs were more clonally expanded (*Figure 5e*). Finally, by calculating the ratio of binding clones to all clones, we could show a clear positive correlation between organ overlap and antigen specificity, where 15–27% of shared clones across all organs could be confirmed as antigen-binding clones (*Figure 5f*).

While our combinatorial scFv yeast display libraries could positively detect a substantial number of antigen-binding clones, it comes with the experimental limitation of random pairing of VH and VL chains. Therefore, in order to reconstruct the natural pairing of VH and VL chains, we performed single-cell antibody repertoire sequencing (10x Genomics, see 'Materials and methods') on IgG+ B cells and plasmablasts/plasma cells from lymphoid organs (bone marrow, spleen, and left axillary and inguinal lymph nodes) of a mouse with a strong humoral immune response from receiving multiple booster immunizations (*Figure 5g*, *Figure 5—figure supplement 3a*). Single-cell antibody sequencing recovered a range of 443–3416 IgG+ B cells per organ (*Supplementary file 4*). Clonal repertoire analysis was performed using both VH and VL chains and clones were defined by identical germline V- and J-genes and 100% a.a. identity of CDRH3-CDRL3. This revealed once again substantial clonal overlap across lymphoid organs, including 26 clones shared across at least three organs and 157 clones shared in at least two organs (*Figure 5h*). Antibody (scFv) expression and binding experiments confirmed that a substantial fraction (33%; 6 out of 18) of tested single-cell clones shared across organs were indeed antigen-specific (*Supplementary file 5* and *Figure 5—figure supplement 3b*).

## Discussion

A central pillar of antibody repertoires is their inherently high diversity (estimated to be >$10^{13}$ in mice) (*Greiff et al., 2017a*). Therefore, each lymphoid organ may be populated with an exclusive repertoire of naive B cells, which following a single immunization will induce independent B-cell activation, diversification, and germinal center reaction events. To investigate how these dynamic processes shape antibody repertoires across lymphoid organs during an immune response, we performed deep sequencing of antibody repertoires using two cohorts of mice that were immunized either once or three times.

Analysis of antibody repertoires in the single immunization cohort revealed extensive clonal expansion in lymph nodes from both left and right sides, suggesting antigen-induced selection and expansion of an IgG-class-switched subset of B cells. These may be comprised of early germinal center B cells and short-lived plasmablasts, the latter being generated within the first days after an immunization or infection and are considered as transient precursors of plasma cells (*Blink et al., 2005*; *Kallies et al., 2004*; *Palm and Henry, 2019*). A recent study reported similar findings of highly expanded germinal center B cells in draining lymph nodes 7 days after influenza infection in mice (*Mathew et al., 2021*). Overlap analysis of the most expanded clones revealed distinct physiological axes: inguinal and axillary lymph nodes from the left side shared the majority of their expanded clones, whereas the same was true for right-side lymph nodes, with minimal overlap across left and right lymph nodes. This may reflect migration of antigen-activated B cells and/or antigen transport across the circulatory network, and lymphatic connections between inguinal and axillary lymph nodes have been well described

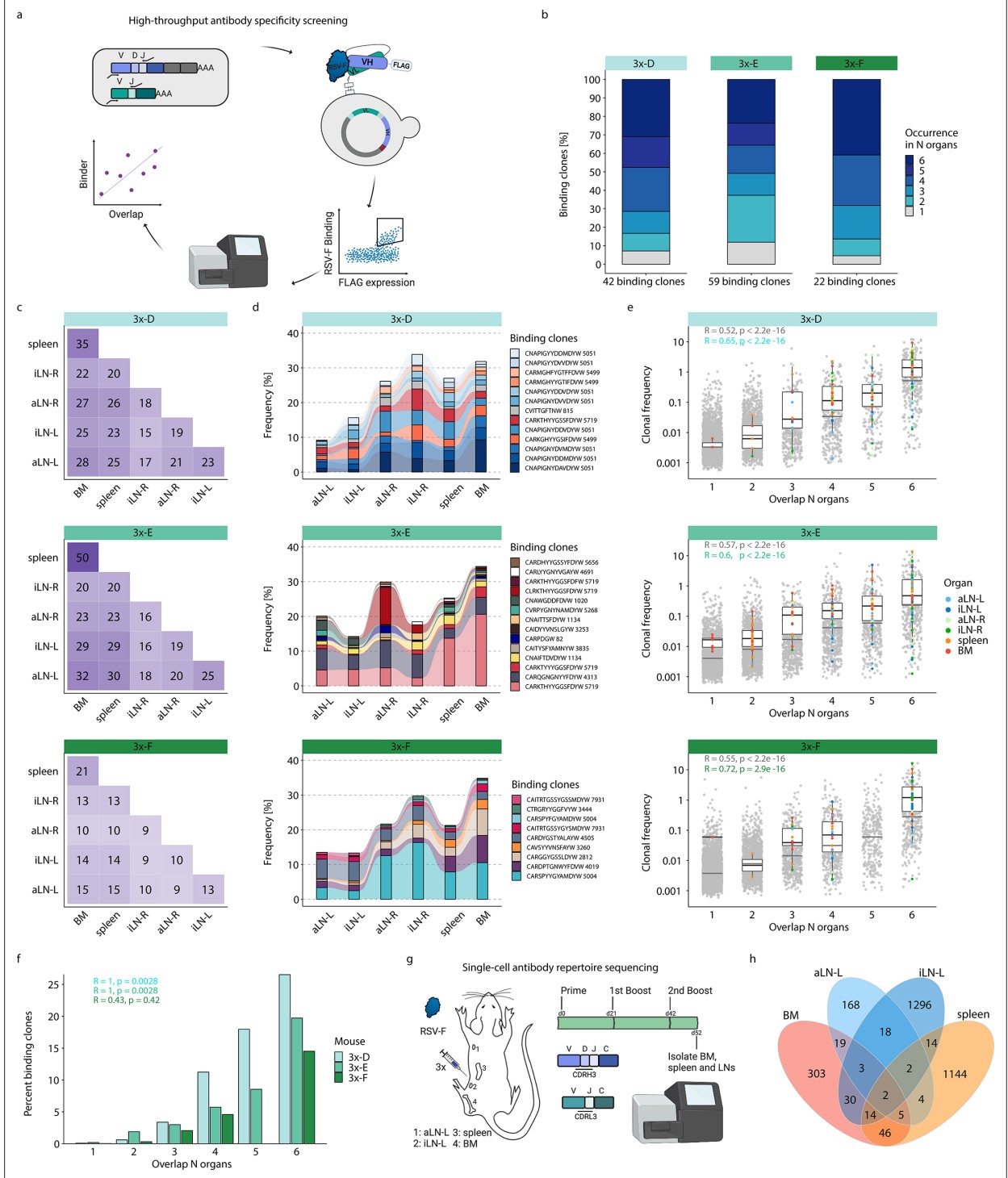

**Figure 5.** High-throughput antibody screening reveals that clonal overlap across lymphoid organs correlates with antigen-specificity. (**a**) Workflow of high-throughput antibody screening by yeast display: antibody (scFv) libraries are generated by combinatorial pairing of variable heavy (VH) and variable light (VL) genes derived from BM of cohort-3x mice and expressed on the surface of yeast cells, followed by isolation of antigen-binding cells by fluorescence-activated cell sorting (FACS) and identification of clones by deep sequencing and repertoire analysis. (**b**) Bar plots represent proportions of binding clones occurring in *N* (1–6) organs within each corresponding mouse of cohort-3x (3x-D, 3x-E, and 3x-F). (**c**) Heatmaps depicting numbers of shared binding clones between two organs within each mouse. (**d**) Tracking of binding clones shared across all six lymphoid organs and their frequencies for each mouse. Legend displays CDRH3 of each clone with the number indicating the corresponding clonotype group. (**e**) Distribution of clonal frequencies per degree of organ overlap (1–6) for all clones (gray) and binding clones (colored). Gray horizontal lines represent the median frequency of all clones, black lines including box plots represent the median frequency of binding clones within the corresponding overlap sections (see

*Figure 5 continued*

color legend for organ origin). R and p-values of Spearman correlation analysis for all clones (gray) and binding clones only (green) are depicted. (**f**) Bar plots displaying ratios of binding clones to all clones with regard to their degree of organ overlap (1–6) for each mouse. R and p-values of Spearman correlation analysis for each mouse are depicted. (**g**) Workflow for RSV-F immunizations into the left lateral flank of one Balb/c mouse, followed by single-cell antibody repertoire sequencing (VH + VL) of lymphoid organs (BM, spleen, aLN-L, iLN-L). (**h**) Venn diagram of shared unique IgG+ clones (based on identical V- and J- genes and CDRH3+CDRL3) across lymphoid organs. BM: bone marrow; aLN-L, -R: left and right axillary lymph nodes; iLN-L, -R: left and right inguinal lymph nodes. Panels (**a**) and (**g**) created with BioRender.com.

The online version of this article includes the following source data and figure supplement(s) for figure 5:

**Figure supplement 1.** High-throughput antigen-specificity screening using scFv yeast display libraries.

**Figure supplement 1—source data 1.** Gel plot of amplified VL and VH genes.

**Figure supplement 1—source data 2.** Gel plot of yeast surface display vector.

**Figure supplement 2.** Tracking of clonotypes across all lymphoid organs.

**Figure supplement 3.** Isolation of B-cell subsets and screening of monoclonal yeast cells expressing single-chain variable fragments (scFvs).

(*Harrell et al., 2008*; *Nham et al., 2012*). A majority of clones that were expanded in lymph nodes were also found in spleen and bone marrow, however, exhibiting lower frequencies. This may indicate early homing of B cells to the spleen and bone marrow where they reside as terminally differentiated plasma cells (*Manz et al., 1997*; *Slifka et al., 1998*). Previous studies examining early B-cell responses in immunized mice have reported the presence of antigen-specific antibody-secreting cells (ASCs) in the spleen and bone marrow as early as 5–10 days after immunization (*Blink et al., 2005*; *Slocombe et al., 2013*; *Takahashi et al., 1998*). Our findings, including observed clustering and phylogenetic tree analysis showing significant transitions from the spleen to the bone marrow in cohort-1x mice, are consistent with these earlier observations and suggest that antigen-specific ASCs can be detected in both organs at the repertoire level 10 days after a single immunization. Performing analysis on all clones, however, revealed limited clonal diversification and low overlap across all lymphoid organs, demonstrating that a primary and weak humoral response is characterized by clonal expansion and segregation of a few clones to particular organs while exhibiting mostly lymphoid organ-exclusive repertoires.

Strikingly, a strong humoral response resulted in a substantial number of shared clones in all mice, exhibiting an up to 70-fold increase in clonal overlap across all six lymphoid organs. Global repertoire similarity measurements confirmed high degrees of pairwise clonal overlap between organs, with shared clones exhibiting similar frequencies. Two mechanisms may be contributing to this high clonal overlap and repertoire similarity in lymphoid organs: (i) subcutaneous injection of soluble antigen results in drainage from interstitial space into lymphatic vessels and is transported directly to lymph nodes (*Reddy et al., 2010*; *Reddy et al., 2006*), with eventual transport into systemic circulation, including the spleen *Irvine et al., 2013*; and (ii) migration of antigen-presenting cells (e.g., dendritic cells) (*Worbs et al., 2017*) and antigen-specific memory B cells via lymphatic and systemic circulation (*Hampton and Chtanova, 2019*). Network-based analysis revealed extensive clonal diversification, which is consistent with the refueling of ongoing germinal center reactions (which usually subside after 30 days) (*Mesin et al., 2020*). This supports antigen transport to all secondary lymphoid organs and B cells, re-entering or still residing in germinal centers, undergoing multiple rounds of selection and affinity maturation, which was reflected by high levels of somatic hypermutation across all organs. Moreover, phylogenetic analysis was performed to investigate if physiological consolidation across lymphoid organs was a consequence of B-cell migration and indeed revealed significant clonal lineage transitions between spleen and bone marrow as well as from spleen to left lymph nodes (side of immunization). Antibody repertoire studies in humans provide evidence for this connection, as shared B-cell clones have been identified across blood-rich tissues including the bone marrow, spleen, lung, and liver (*Meng et al., 2017*; *Domínguez Conde et al., 2022*). Performing three rounds of immunization with a 21-day interval allows for the reactivation of memory B cells present in secondary lymphoid organs and provides sufficient time for LLPCs to home to the bone marrow, which represents a physiological niche that supports long-term residency of LLPCs and immunological memory (*Manz et al., 1997*; *Slifka et al., 1995* and *Slifka et al., 1998*). However, recent studies have challenged the conventional view that all LLPCs exclusively reside in the bone marrow; instead, these cells may recirculate between multiple bone marrow niches and can even migrate to the spleen, suggesting the presence

of a shared reservoir of LLPCs throughout the body (*Aaron and Fooksman, 2022*; *Benet et al., 2021*). This dynamic behavior may contribute to the observed transitions in both directions between spleen and bone marrow as well as the high degree of repertoire consolidation observed across all lymphoid organs in cohort-3x mice.

By performing high-throughput screening using yeast surface display, we confirmed that a major fraction of shared clones across organs was antigen-specific. Also, we observed that spleen and bone marrow exhibited the highest repertoire similarity by sharing the majority of their detected binding clones. Moreover, we uncovered that shared binding clones displayed relatively high clonal frequencies, which is in line with reports showing that antigen-specific serum titer responses are mainly a result of a few expanded B-cell clones, reflecting oligoclonal B-cell responses upon vaccination or infection (*Lavinder et al., 2014*; *Reddy et al., 2010*; *Voss et al., 2021*).

Here, we have systematically uncovered that strong humoral responses induce a high degree of physiological consolidation of antibody repertoires and antigen-binding clones, observed across multiple distinct lymphoid organs, which is also associated with an overall reduced clonal diversity. Although these findings are dependent on immunization parameters (i.e., antigen delivery [viral-based, protein, mRNA] and adjuvant types), they provide important implications for vaccinology. For example, multiple immunization schemes are often employed clinically (e.g., mRNA vaccinations for Covid-19 and hepatitis B subunit vaccines) (*Lederer et al., 2020*; *West and Calandra, 1996*) to induce strong humoral responses (based on antigen-binding serum titers); however, this may come at the expense of reducing antibody repertoire diversity across lymphoid organs. Future vaccine strategies may benefit by attempting to balance the strength of the induced humoral response with the physiological landscape of clonal repertoire diversity, as more diverse antibody repertoires in lymphoid organs may facilitate greater breadth of antigen coverage and perhaps enhance resistance to viral escape mutations (e.g., SARS-CoV-2 and influenza).

## Materials and methods

### Mouse immunizations and lymphoid organ harvest

All mouse experiments were performed under the guidelines and protocols approved by the Basel-Stadt cantonal veterinary office (Basel-Stadt Kantonales Veterinäramt, Tierversuchsbewilligung #2582). In total, seven female Balb/c mice (BALB/cJRj; Janvier Laboratories France, 10 weeks old) were housed under specific pathogen-free conditions and maintained on a standard chow diet.

Mice were randomly allocated into two cohorts of three mice and immunized subcutaneously into the left flank with 10 µg respiratory syncytial virus (pre-)fusion glycoprotein (RSV-F, expressed and purified as previously described [*Sesterhenn et al., 2019*]) in combination with 50 µg poly(I:C) adjuvant (HMW, InvivoGen tlrl-plc) per mouse. RSV-F protein was resuspended in 1× DPBS (Gibco 14190144) together with poly(I:C) adjuvant to a final injection volume of 100 µl per mouse shortly before each immunization. One cohort received a single immunization (cohort-1x), whereas the other cohort received three immunizations with 21-day intervals between booster immunizations (cohort-3x). Mice from both cohorts were euthanized by $CO_2$ asphyxiation and cervical dislocation 10 days after the single or third immunization, respectively. From each mouse, left and right axillary lymph nodes, left and right inguinal lymph nodes, spleen, and both femurs and tibiae were harvested. In order to avoid contamination, pipettes and scissors were cleaned and sterilized after each organ isolation step. Blood samples were collected by cardiac puncture, followed by separation of serum after 30 min clotting at room temperature and two subsequent centrifugation steps at $6000 \times g$ for 15 min. Lymph nodes and spleens were directly placed in 1–2 ml RNAlater (Sigma R0901) and stored at 4°C for up to 6 days and transferred to –20°C until further processing. Isolated femurs and tibiae were stored in a buffer (PBS, containing 0.5% BSA [Miltenyi Biotec 130-091-376] and 2 mM EDTA [Thermo Fisher Scientific AM9260G] on ice). For bone marrow isolation, both ends of femurs and tibiae were cut with surgical scissors and bone marrow was extracted by flushing 3–5 ml sterile and ice-cold buffer through the bones using a 26G needle (BD Microlance 3 300300). The isolated bone marrow was then filtered through a 40 µM nylon cell strainer (FALCON 352340) and cell-suspension was centrifuged at $300 \times g$ for 10 min at 4°C. The supernatant was removed and 1 ml of Trizol (Life Technologies 15596) was used to resuspend samples before storing at –80°C until further processing.

For comparison of immunized IgG+ and untreated IgM+ antibody repertoires, we used three untreated female Balb/c mice (BALB/cJRj; Janvier Laboratories France, 8 weeks old), which were housed together under specific pathogen-free conditions and maintained on a standard chow diet. After an acclimatization period of 17 days upon arrival, mice were euthanized by $CO_2$ asphyxiation and cervical dislocation followed by isolation of left inguinal lymph node, spleen, and both femurs and tibias and organs were processed as described above.

For the single-cell sequencing experiment, the same procedure as for cohort-3x mice was done for one female Balb/c mouse (BALB/cJRj; Janvier Laboratories France, 9 weeks old). At the end of the experiment, left-side axillary and inguinal lymph nodes, as well as spleen and both hindlegs were isolated and stored in freshly prepared and ice-cold buffer (PBS +2% fetal bovine serum [FBS] [Thermo 16250078] + 2 mM EDTA, sterile filtered). Bone marrow was isolated as described above and kept on ice until further processing for FACS.

## RNA isolation from lymphoid organs

Mouse spleens and lymph nodes were removed from RNAlater solution and transferred to gentle-MACS M tubes (Miltenyi Biotec 130-093-236) and 2 ml microtubes (OMNI 19-629-3, 1.4 and 2.8 mm Ceramic mix) containing 1.5 ml and 0.8 ml ice-cold Trizol, respectively. Spleens and lymph nodes were dissociated using gentleMACS Octo Dissociator (Miltenyi Biotec 130-095-937, RNA_01_01 pre-programmed setting was performed twice) and Bead Ruptor 24 Elite (OMNI 19-040E, 1 × 10 s, 4 m/s, 1 cycle), respectively. Cell suspensions were removed and stored at –80°C (spleen) or subsequently processed for RNA extraction (lymph nodes). All samples were thawed on ice prior to RNA extraction using the PureLink RNA Mini Kit (Life Technologies 12183018A) following the manufacturer's guidelines. Briefly, 0.7 ml of lymph node cell suspension and 1 ml of spleen and bone marrow samples were loaded with chloroform onto 5PRIME Phase Lock Gel Heavy tubes (Quantabio 2302830) for better separation of phases. All other steps were performed according to the manual's instructions (protocol: Using Trizol Reagent with PureLink RNA Mini Kit). Samples were eluted in 30/80 μl water (lymph nodes/spleen, bone marrow) and stored at –80°C until further processing. After RNA isolation, overall RNA quality was determined by RNA ScreenTape Analysis (Agilent 5067-5576), with all samples showing RIN values between 8 and 10.

## Antibody repertoire library preparation and deep sequencing

Antibody VH chain libraries for deep sequencing were constructed using a previously established protocol of molecular amplification fingerprinting (MAF) that incorporates unique molecular identifiers for error and bias correction (*Khan et al., 2016*). Briefly, first-strand cDNA synthesis was performed using Maxima reverse transcriptase (Life Technologies EP0742) following the manufacturer's instructions, using 5 μg total RNA and a gene-specific primer corresponding to constant heavy region 1 of IgG subtypes (IgG1, IgG2a, IgG2b, IgG2c, and IgG3) with an overhang region containing a reverse unique molecular identifier (RID).

After cDNA synthesis, samples were subjected to a left-sided 0.8× SPRIselect bead cleanup (Beckman Coulter B23318). Quantification of target-specific cDNA by a digital droplet (dd)PCR allowed exact input of first-strand cDNA copies of each sample into the next PCR reaction. Multiplex PCR was performed using a forward primer set annealing to framework 1 regions of VH including an overhang region of forward unique molecular identifier (FID) and a partial Illumina adapter as well as a reverse primer containing a partial Illumina sequencing adapter. The reaction mixtures contained 135,000 first-strand cDNA copies, 15,000 synthetic spike-ins (for quality control), primer mix, and 1× KAPA HiFi HotStart Uracil+ReadyMix (KAPA Biosystems KK2802) with the Uracil+ version enabling efficient high-fidelity amplification of multiplex primers containing deoxyinosine. PCR reactions were cleaned using left-sided 0.8× SPRIselect bead clean-up as before and product was quantified again using ddPCR assay. Finally, an Illumina adaptor-extension singleplex PCR step was performed using 820,000 copies of the previous PCR product with 1× KAPA HiFi HotStart ReadyMix (KAPA Biosystems KK2601), followed by double-sided (0.5×–0.8×) SPRIselect bead cleanup and sample elution in Tris-EDTA buffer (Fluka 93302-100ML). Overall library quality and concentration were determined on the Fragment Analyzer (Agilent DNF-473 NHS Fragment kit). Libraries were then pooled in Tris-HCl + 0.1% Tween-20 (Teknova T7724) and sequenced on an Illumina MiSeq using the reagent v3 kit (2 × 300 bp) with 10% PhiX DNA for quality purposes. All 36 IgG+ samples were processed and sequenced

in three batches of 12 samples by preparing sequencing libraries of one mouse per cohort per batch with mixed sample order.

For sequencing of IgM VH libraries of untreated mice, the same workflow was used with the following modifications (based on *Greiff et al., 2017a*):

> Primer for cDNA synthesis: TTGGCACCCGAGAATTCCACTGHHHHHHACAHHHHHHACAHH HHNATTCCATGGCCACCAGATTCTT
> Probe for ddPCR of IgM amplicons: /56-FAM/ CC +AA + AT +GT + CTT +CCC/3IABkFQ/
> Primer for IgM J region amplification: GTC TC/ideoxyI/ /ideoxyI/CA GAG AGT CAG

## Antibody repertoire analysis

Raw FASTQ files were processed using a custom CLC Genomics Workbench 10 script. First, low-quality nucleotides were removed using the quality trimming option with a quality limit of 0.05, followed by merging of forward and reverse read pairs and removal of sequences not aligning to mouse IGH constant sequences. Only amplicons between 350 and 600 base pairs were kept for further processing. Preprocessed sequences were then used as input for the previously established MAF bioinformatic pipeline 12 (*Khan et al., 2016*) to perform alignment of sequences as well as error and bias correction (see *Supplementary file 1* for cohort-1x and cohort-3x mice; for untreated Balb/c mice 80,000 preprocessed sequences per organ were used). For all downstream analysis, an output TSV file ('CDR3_Tot_Table') for each sample was used that listed each unique CDRH3 of the consensus-built data per row, including majority V-gene, majority J-gene information, and corrected MAF frequency information. Within the study, we defined sequences with identical CDRH3 a.a. sequence and matching V-, and J-genes to be a clone. For investigation of clonally related B-cell variants (here referred to as clonotypes), we performed hierarchical clustering (single linkage) of antibody sequences with identical V-, and J-gene and matching CDRH3 a.a. lengths and >90% a.a. CDRH3 identity (*Friedensohn et al., 2018*).

For phylogenetic tree analysis, we used the output TSV file 'Polished_Annotated_Table' containing all corresponding consensus built and error-corrected read sequences belonging to a clone (here referred to as clonal sequences) in order to obtain intraclonal variants for building lineage trees.

All downstream analyses were carried out using customized R scripts in R version 4.0.3 (*R Development Core Team, 2020*).

## Comparative analysis of germline V-gene usage

Counts for unique germline V-genes were summarized for all unique clones within each sample (without including clonal frequency information) and Pearson's correlation coefficients were calculated between two repertoires across all samples.

## Quantification of clonal expansion

To assess the distribution of clonal frequencies in antibody repertoires, we employed Hill-based evenness profiles as previously described (*Greiff et al., 2015*). Hill-based evenness profiles are adapted from mathematical ecology and offer a method to analyze the state and uniformity of clonal expansion using a continuum of diversity indices (*Greiff et al., 2015*; *Hill, 1973*). This allows for the comparison of clonal frequency distributions across antibody repertoires.

Hill-based evenness is defined as $^{\alpha}E = {^{\alpha}D}/SR$, where $^{\alpha}D$ represents the Hill-based diversity profile and SR (species richness) the number of all clones present in the specific repertoire.

Hill-based diversity ($^{\alpha}D$) is calculated as follows:

$$^{\alpha}\mathrm{D} = \left( \sum_{i=1}^{n} f_i^{\alpha} \right)^{\frac{1}{1-\alpha}}$$

with $f$ being the clonal frequency distribution, $f_i$ being the frequency of each clone, and $n$ the total number of clones. The parameter $\alpha$ determines the impact of high-frequency clones. For instance, $\alpha = 0$ indicates the number of unique clones in a repertoire without weighting clones (thus resembling SR), while with increasing values of parameter $\alpha$, high-frequency clones are given more weight. Shannon evenness with $\alpha = 1$ is commonly utilized to assess clonal expansion of immune repertoires (*Amoriello*

*et al., 2021*; *Csepregi et al., 2020*; *Galson et al., 2015*; *Greiff et al., 2015*; *Greiff et al., 2017a*; *Jost, 2006*; *Rosenfeld et al., 2018*; *Venturi et al., 2007*). Values range from 0 to 1, with values closer to 1 indicating a uniform frequency distribution of clones, while values closer to 0 indicating the prevalence of one or a few highly expanded clones in a polarized repertoire.

## Assessment of repertoire similarities

Analysis of pairwise repertoire similarities was performed by quantification of pairwise Jaccard and cosine similarity using R package immunarch 0.6.5 (*Nazarov et al., 2021*). Briefly, we calculated the pairwise Jaccard index by quantifying the size of intersection between two repertoires A and B and dividing it by the length of the union of the same repertoires to assess repertoire similarities based on clone overlap across all samples:

$$J\left(A,B\right) = \frac{|A \cap B|}{|A| + |B| - |A \cap B|}$$

Cosine similarity measurements can be used to indicate the degree of antibody repertoire similarity between two repertoires A and B by including both clonal overlap and frequency information (by giving weight to clonal frequencies), ranging from 0 to 1, thus higher cosine values correspond to higher similarities between two repertoires (*Rosenfeld et al., 2018*):

$$similarity(A,B) = \frac{\sum_{i=1}^{n} A_i B_i}{\sqrt{\sum_{i=1}^{n} A_i^2} \sqrt{\sum_{i=1}^{n} B_i^2}}$$

Additionally, overlapping unique clones were represented graphically via Venn diagrams per mouse and cohort using R package venn 1.9 (*Dusa, 2020*).

## Network-based analysis

Sequence-similarity networks of clonotypes were constructed as described in the figure legends. Briefly, each node represents a unique clone that is connected by edges to clones with a Hamming distance of 1 between CDRH3 a.a. (edit distances were calculated using the R package stringdist v.0.9.6.3; *van der Loo et al., 2021*). The relative size of each node was proportional to its clonal frequency (in log and rescaled for each mouse). Networks were visualized with the Kamada–Kawai layout; all network and connectivity plots were generated using the R package igraph v.1.2.6 (*Csardi and Nepusz, 2006*).

## Phylogenetic tree analysis

Phylogenetic analysis of BCR sequences is best performed using full-length BCR sequences. Full-length V region sequences were obtained from MAF output TSV file 'Polished_Annotated_Table'. Only sequences that were productively rearranged, associated with sequences in the final CDRH3 dataset, did not correspond to spike-in control sequences and contained at most one missing ('N') nucleotide were retained for final analysis. Within each sample, identical sequences using the same primers were collapsed and sequences with less than 3 associated reads were discarded using *presto* v0.6.2 (*Vander Heiden et al., 2014*).

Accurate estimation of SHM requires an appropriate Ig germline reference database. BALB/c IGHV germline reference sequences (*Jackson et al., 2022*) were obtained from OGRDB (*Lees et al., 2020*) on 29 April 2022. Because BALB/c IGHJ and IGHD sequences were not available from OGRDB (and were not used to calculate somatic hypermutation), IGHD and IGHJ sequences from the IMGT GENE-DB v3.1.24 (*Giudicelli et al., 2005*) mouse immunoglobulin database were used. Sequences were aligned to this combined immunoglobulin database using IgBlast v1.18.0 (*Ye et al., 2013*). The level of SHM for each sequence was initially calculated as its length-normalized Hamming distance from its predicted germline sequence along the IGHV region (IMGT positions 1–312) using *shazam* v1.1.0 (*Yaari et al., 2013*). Incorrect germline reference sequences are expected to result in a higher estimated level of SHM. To assess the accuracy of our germline reference database, we compared the level of SHM in each sequence when aligned to the BALB/c IGHV reference database and the IMGT

IGHV mouse reference database. Only 1.2% of sequences had a higher SHM level when aligned to the BALB/c database. These possibly represented alleles not included in the published BALB/c germline database and were removed. By contrast, 24.4% of sequences had lower level of SHM when aligned to the BALB/c database, confirming its accuracy.

Before building phylogenetic trees, it is important to identify clonal clusters, which represent B cells that descend from a common IGHV, D, and J rearrangement. Sequences from all tissue samples within each mouse were clustered into clones by partitioning based on common IGHV gene annotations, IGHJ gene annotations, and junction lengths. Within these groups, sequences differing from one another by a length normalized amino acid Hamming distance of 0.1 within the CDRH3 were defined as clonotypes by single-linkage clustering using *scoper* v1.2.0 (**Nouri and Kleinstein, 2018**). This Hamming distance threshold and use of amino acid Hamming distance were used for consistency of clonotype definition in other analyses within this manuscript. Within each clonal cluster, germline sequences were reconstructed with D segment and N/P regions masked (replaced with 'N' nucleotides) using the createGermlines function within *dowser* v1.0.0 (**Hoehn et al., 2022**).

Phylogenetic analysis was performed using *dowser* v1.0.1 (**Hoehn et al., 2022**). Within each mouse, sequences from the same tissue differing only by ambiguous nucleotides were collapsed. For computational efficiency, large clones were randomly down-sampled to a maximum size of 100 sequences. Clones containing sequences from only one organ, or fewer than 10 sequences total, were removed. Lineage tree topologies and branch lengths were estimated using *IgPhyML* v1.1.4 (**Hoehn et al., 2019**) and visualized using *ggtree* v3.0.4 (**Yu et al., 2017**). The SP permutation test (**Hoehn et al., 2022**) was used to assess whether lineage trees showed signs of migration among organs in a particular direction. Briefly, given the set of organ labels at the tips of each phylogenetic tree, a maximum parsimony algorithm implemented in *IgPhyML* was used to reconstruct the set of internal node organ labels, resulting in the fewest number of organ changes along the tree. For all trees within each mouse, the number and direction of organ changes along all trees were recorded and normalized by the total number of changes to give the SP statistic. Organ states were then randomized within each tree, and the resulting SP statistic was calculated for these permuted trees. The difference between observed and permuted SP statistics ($\delta$) was recorded, and this process was repeated for 1000 replicates. The p-value for enrichment of changes between tissues (i.e., $\delta > 0$) is the proportion of replicates in which $\delta \leq 0$. If p<0.05 and $\delta > 0$ for a given pair of tissues, this indicates a significantly more biased ancestor/descendant relationship from one tissue to the other than expected by chance in the lineages surveyed. Tree topologies were not re-estimated throughout permutation replicates; however, to control the false-positive rate of the SP test, all trees were randomly down-sampled to a maximum tip-to-state change ratio of 20 for each repetition. Clusters of internal nodes separated by zero-length branches (polytomies) were reordered using nearest-neighbor interchange moves to minimize the number of changes along the tree and appropriately represent possible directions of migration. For analysis in *Figure 4—figure supplement 1c*, the SP test was repeated as above but node organ labels were permuted among trees within each mouse. Further, tissue changes were quantified in either direction among tissues. This modified SP test more directly quantifies association among organs, rather than biased ancestor–descendant relationships. The scripts for reproducing these analyses are available at https://bitbucket.org/kleinstein/projects.

## Antigen ELISA

Standard antigen ELISAs were performed to measure mice sera for RSV-F-specific titers. High binding 96-well plates (Costar CLS3590) were coated overnight with 2 µg/ml of RSV-F in PBS at 4°C. Plates were washed three times in PBS containing 0.05% (v/v) Tween-20 (AppliChem A1389) (PBST) and blocked for 2 hr at room temperature with PBS containing 2% (m/v) non-fat dried milk powder (AppliChem A0830) and 0.05% (v/v) Tween-20 (PBSM). After blocking, plates were washed three times with PBST. Sera were prediluted depending on cohort (1:150 for cohort-1x and 1:450 for cohort-3x) and then serially diluted in 1:3 steps in PBSM across the plate. For normalization of multiple plates, motavizumab, a humanized monoclonal RSV-F binding antibody, was used as a standard curve by serial 1:5 dilutions (starting concentration: 1 µg/ml). Plates were incubated for 1 hr at room temperature and washed three times with PBST. HRP-conjugated rat monoclonal (187.1) anti-mouse kappa light chain antibody (abcam ab99617, 1:1500 dilution in PBSM) and HRP-conjugated goat polyclonal anti-human IgG F(ab')$_2$ fragment antibodies (Jackson ImmunoResearch 109-036-008, 1:5000 dilution in PBSM)

were used as secondary detection antibodies for sera and motavizumab, respectively. Plates were incubated at room temperature for 1 hr again, followed by three washing steps with PBST. ELISA detection was performed using the 1-Step Ultra TMB-ELISA Substrate Solution (Thermo 34028) and reaction was terminated adding 1 M $H_2SO_4$. Absorption at 450 nm was measured with the Infinite M200 PRO NanoQuant (Tecan) and data were analyzed using Prism V7 (GraphPad). Serum endpoint titer was determined using the last diluted specimen that gave a positive signal compared to the negative control (no serum).

## Cloning and expression of yeast display antibody libraries

For each mouse of cohort-3x, 2–3 independent combinatorial scFv yeast display libraries were constructed using corresponding bone marrow RNA samples for the detection of RSV-F binding antibody sequences (based on CDRH3). It should be noted that constructing a randomly paired VH VL library may not cover the whole repertoire due to improper VH VL pairing.

Briefly, reverse transcription of bone marrow RNA samples (same as used for antibody repertoire sequencing) was performed using Maxima reverse transcriptase (Life Technologies EP0742) following the manufacturer's instructions. Two 40 µl reactions per sample were prepared using 1 µg total RNA and IgG gene-specific primers (p1/p2, see *Supplementary file 6*) for heavy chains or oligo(dT)$_{18}$ primer (Thermo, SO131) for light chains. After cDNA synthesis, split-pool PCR was performed for VH and VL samples by setting up eight parallel 25 µl reactions for each sample, containing 4 µl of first-strand cDNA, the VH or VL primer mix (reported in *Supplementary file 6*, final concentration of 0.5 µM each) and KAPA HiFi HotStart Uracil+ReadyMix (Kapa Biosystems KK2802) or KAPA HiFi HotStart ReadyMix (Kapa Biosystems KK2601) for VH and VL reactions, respectively. The following cycling conditions were used for VH amplification: initial denaturation 3 min at 95°C, 30 cycles with denaturation at 98°C (20 s), annealing at 60°C (30 s), elongation at 72°C (45 s), and final elongation at 72°C (1 min); for VL amplification: initial denaturation 3 min at 95°C, 30 cycles with denaturation at 98°C (20 s), annealing at 54°C (15 s), elongation at 72°C (30 s), and final elongation at 72°C (1 min). After split-pool PCR, the corresponding eight reactions were pooled and subjected to a first PCR clean-up with QIAquick PCR Purification Kit (QIAGEN 28016) to concentrate samples, followed by a final clean-up step by agarose-gel extraction (2% [w/v] gel) with Zymoclean Gel DNA Recovery kit (Zymo Research D4001). Yeast scFv display libraries were generated using the amplified VH and VL DNA libraries from above and the pYD1 yeast surface display vector (Addgene plasmid #73447; https://www.addgene.org/73447/; RRID:Addgene_73447) (*Kieke et al., 1997*), which was modified to include a BamHI restriction site between HA- and FLAG-epitope tags (from now on referred to as 'pYD1-BamHI') [containing galactose (GAL)-induced GAL1 promoter and a BamHI restriction site in order to insert both VH VL libraries separated by a (Gly$_4$Ser)$_3$ linker as scFv format] (*Figure 5—figure supplement 1b*). All in-frame scFv sequences resulted in a C-terminal FLAG-tag to identify scFv-expressing yeast cells. The vector was linearized using BamHI-HF (NEB R0136S) in Cutsmart buffer (NEB B7204S) by incubation for 45 min at 37°C and immediate purification using Zymo DNA Clean & Concentrator kit (Zymo D4005). Shortly before transformation, inserts and cut vector were dialysed for 30 min using 0.025 µm MF-Millipore membrane filters (Merck VSWP01300), and purity of final products was confirmed by agarose gels and concentration measured using NanoDrop 2000c (Thermo Scientific ND-2000). Next, 8–10 separate reactions per library were prepared by combining 1 µg of linearized vector with 1 µg of each VH and VL inserts for co-transformation into 330 µl competent EBY100 yeast cells. Electroporation of EBY100 yeast cells was performed with Bio-Rad MicroPulser Electroporator (Bio-Rad 1652100) using pre-chilled Gene Pulser cuvettes (2 mm gap, Bio-Rad 1652086) as described previously (*Benatuil et al., 2010*; *Boder and Wittrup, 1997*). Transformation efficiency was determined by plating tenfold serially diluted yeast cells on SD-CAA + 2% glucose plates following incubation at 30°C for 48 hr. This workflow resulted in ~1–3 × 10$^6$ transformants per library. Transformation efficiency could be optimized using longer homology overhangs in the primers used for VH and VL amplification. As positive control, monoclonal yeast cells expressing the RSV-F binding monoclonal antibody palivizumab as scFv format were generated (see 'Generation and screening of monoclonal antibodies by yeast display').

## Screening yeast display antibody libraries for antigen binding by FACS

Yeast cells were cultured in yeast nitrogen base-casamino acids (YNB-CAA) (BD Biosciences 223120) + 2% D-(+)-glucose (Sigma G5767-500G) growth medium including phosphate buffer (5.4 g/l $Na_2HPO_4$, 8.6 g/l $NaH_2PO_4 \cdot H_2O$) and 1× Penicillin-Streptomycin (P/S) (Life Technologies 15140122) to avoid bacterial contamination. A day before sorting, a fraction of yeast cells, >10-fold larger than the (initial or sorted) library sizes, were pelleted at 3.000 × $g$ for 3 min and induced by resuspension in YNB-CAA + 2% D-(+)-galactose (Sigma G0625-500G) + phosphate buffer + P/S induction medium at a final $OD_{600}$ of 0.5. Cells were grown at 30°C overnight, with shaking at 250 rpm. On the day of sorting, a total of 10–50-fold larger than the (initial or sorted) library sizes were pelleted at 7.000 × $g$ for 2 min at 4°C, washed twice in buffer (PBS + 0.5% BSA) and stained with the anti-FLAG-phycoerythrin (PE) antibody (anti-DYKDDDDK Tag, RRID:AB_2563148; BioLegend 637310) for confirmation of scFv expression and RSV-F protein conjugated with Alexa Fluor 647 (AF647) to select antigen-binding cells. For the antibody labeling steps, yeast cells were resuspended in ice-cold buffer containing 0.02 µg/µl anti-FLAG-PE and 0.01 µg/µl RSV-F-AF647 with a final cell density of ~$10^5$ cells/µl. Cells were incubated for 30 min at 4°C with shaking at 450 rpm. Finally, cells were washed twice and filtered before flow sorting. The following machines were used: Sony SH800S (Sony Biotechnology), Sony MA900 (Sony Biotechnology), or BD FACS Aria III (BD Biosciences). Flow cytometry data was analyzed using FlowJo V10.4.2 (FlowJo, LLC).

For setting the sort gates, monoclonal yeast cells expressing palivizumab as scFv were used as positive control and corresponding libraries stained with anti-FLAG-PE only were used as negative controls. Cells were sorted for double-positive cells (AF647+/PE+), and recovered yeast cells were cultured for expansion in glucose growth medium. The expansion, induction, staining, and recovery were performed three additional times with the same antigen and working concentration. Gates were set less stringent due to the low fraction (~0.1–0.3%) of double-positive cells for the first sort, but stringency was increased with further sort rounds. After the final sort, the enriched cells were screened for purity of double-positive yeast populations using BD LSR Fortessa (BD Biosciences) and enriched yeast plasmid libraries were extracted using Zymoprep Yeast Plasmid Miniprep II kit according to the manufacturer's instructions (Zymo Research D2004).

## Deep sequencing of antigen-binding yeast display libraries

The purified plasmid DNA was used as a template for PCR using a primer pair (p50 and p51, see *Supplementary file 6*) targeting the VH of the recombined scFv insert for detection of RSV-F binding antibody sequences (based on CDRH3). Briefly, per library two 25 µl reactions were prepared using 2 µl of plasmid template each together with p50 and p51 primers (final concentration of 0.8 µM) and KAPA HiFi HotStart ReadyMix. The following cycling conditions were used: initial denaturation 3 min at 95°C, 23 cycles with denaturation at 98°C (20 s), annealing at 65°C (30 s), elongation at 72°C (45 s), and final elongation at 72°C (1 min). Samples were subjected to a left-sided 0.8× SPRIselect bead cleanup and an Illumina adaptor-extension PCR step was performed using 150 ng of purified PCR product into 50 µl singleplex PCR with 1× KAPA HiFi HotStart ReadyMix and Illumina Nextera primers for dual indexing, followed by double-sided (0.5×–0.8×) SPRIselect bead cleanup and sample elution in Tris-EDTA buffer. Overall library quality and concentration was determined on the Fragment Analyzer (Agilent DNF-473 NHS Fragment kit). All libraries were then pooled and sequenced on an Illumina MiSeq using the reagent v3 kit (2 × 300 bp) with 20% PhiX DNA for quality purposes.

Raw sequencing files obtained from Illumina MiSeq were aligned using the 'analyze amplicon' command of the MiXCR software package (*Nazarov et al., 2015*). For downstream analysis, sequences with a clone count of at least 2 were kept and all unique CDRH3s obtained from libraries of the corresponding mouse were pooled. This CDRH3 pool was then matched with the corresponding mouse repertoire data of all six lymphoid organs in order to assign RSV-F specificity to antibody clones (referred to as 'binding clones').

## FACS isolation of B-cell subsets

We performed FACS to isolate IgG+ CD19+ B cells from lymph nodes and spleen as well as plasma cells and plasmablasts (CD19+, TACI+, CD138+) from spleen and bone marrow (as previously described, *Pracht et al., 2017*) in order to generate antibody repertoire libraries for single-cell sequencing.

First, single-cell suspensions were prepared from spleen, left axillary, and left inguinal lymph nodes by transferring each organ in ice-cold buffer (DPBS, 2% FBS, and 2 mM EDTA) onto a 40 µM nylon cell strainer and softly mashing the organ through the cell strainer into a Petri dish using the plunger tip of a 1 ml syringe (BD Plastipak 303172). Cell suspensions were filtered several times and filters were rinsed with fresh buffer to obtain a homogeneous single-cell suspension. All steps were performed on ice if not stated otherwise, and all centrifugation steps were performed at 300 × *g* for 5 min at 4°C.

For spleen/bone marrow suspensions, red blood cell lysis was performed after centrifugation of cell suspensions, followed by resuspension and incubation in 3/5 ml RBC lysis buffer (eBioscience 00-4333-57) for 5 min at room temperature. Lysis was stopped by adding 10 ml of ice-cold buffer and filtering of cell suspension through a 40 µM nylon cell strainer. 5E7 cells per spleen and bone marrow, and 5-8E6 cells per lymph nodes were used for each staining. To avoid nonspecific binding, cell pellets were resuspended in ice-cold buffer containing unlabeled purified rat anti-mouse CD16/CD32 mAb (Mouse BD Fc Block, BD Pharmingen 553141), 1:50 in PBS + 2% FCS and blocked for 15 min on ice. Next, cells were washed in ice-cold buffer and centrifuged before staining. The pellet was then resuspended to a final concentration of 5E6 cells/ml in buffer containing the respective fluorochrome-coupled antibodies and incubated for 30 min on ice in the dark. The following antibodies were used: Brilliant Violet 421 anti-mouse IgD antibody (RRID:AB_2562743; BioLegend 405725, 1:100 dilution), Brilliant Violet 421 anti-mouse IgM antibody (RRID:AB_10899576; BioLegend 406517, 1:100 dilution), APC-Cyanine7 anti-mouse CD4 antibody (RRID:AB_312699; BioLegend 100413, 1:200 dilution), APC-Cyanine7 anti-mouse CD8a antibody (RRID:AB_312753; BioLegend 100713, 1:200 dilution), APC-Cyanine7 anti-mouse NK-1.1 antibody (RRID:AB_830870; BioLegend 108723, 1:200 dilution), APC-Cyanine7 anti-mouse TER-119 (RRID:AB_2137788; BioLegend 116223, 1:200 dilution), PE-Cyanine7 anti-mouse CD19 (RRID:AB_313655; BioLegend 115519, 1:200 dilution), FITC anti-mouse IgG1 (RRID:AB_493293; BioLegend 406605, 1:100 dilution), FITC anti-mouse IgG2a/2b (RRID:AB_394837; BD Pharmingen 553399, 1:100 dilution), FITC anti-mouse IgG3 (RRID:AB_394840; BD Pharmingen 553403, 1:100 dilution), PerCP-Cyanine5.5 anti-mouse CD138 (RRID:AB_2561601; BioLegend 142509, 1:200 dilution), and PE anti-mouse CD267 (TACI) (RRID:AB_2203542; BioLegend 133403, 1:200 dilution). Cells were incubated with the antibody mix including LIVE/DEAD Fixable Near-IR Dead Cell Stain (1:10,000, Life Technologies L34975) for 30 min at 4°C. Cells were washed twice before cell sorting with FACSAria.

## Single-cell sequencing and analysis of antibody repertoires

Single-cell sequencing libraries were prepared from sorted cells according to the provided 10x Genomics' protocol ('Direct target enrichment – Chromium Single Cell V(D)J Reagent Kits' [CG000166 REV A]). Briefly, sorted single cells were co-encapsulated with gel beads (10x Genomics 1000006) using four lanes of one chromium single-cell chip by loading 1.474–13.000 cells per reaction. V(D)J library preparation was performed using the Chromium Single Cell 5' Library Kit (10x Genomics 1000006) and the Chromium Single Cell V(D)J Enrichment Kit, Mouse B Cell (10x Genomics 1000072) according to the manufacturer's manual. Final libraries were pooled and sequenced on the Illumina NextSeq 500 platform (mid output, 151 cycles, paired-end reads) using an input concentration of 1.6 pM with 1% PhiX.

Raw sequencing files from multiple corresponding Illumina sequencing lanes were merged and BCR sequence reads were processed using Cell Ranger version 3.1.0 (10x Genomics, with murine V(D)J reference GRCm38 [mm10]). Only cells with productive heavy and light chain sequences were retained; in case there was more than one productive heavy or light chain present per cell, the most abundant sequence was kept. Cells with IgG subtypes were used for overlap analysis and clones were defined based on V-, J-gene identity and identical combinations of CDRH3 + CDRL3 amino acid sequences.

## Generation and screening of monoclonal antibodies by yeast display

For the detection of antigen-binding clones with known natural pairing of VH and VL chains, we used organ-shared antibody sequences obtained by single-cell antibody repertoire sequencing. We expressed a subset of the 26 overlapping IgG+ antibody sequences that were at least present in three organs. Clonal grouping of the 26 IgG+ antibody sequences resulted in 17 clonal groups out of which at least one variant was chosen to be expressed as scFv. The selection criteria for choosing sequences

were based on their prevalence across most organs and highest abundance in the bone marrow within the corresponding clonal group. Briefly, full-length sequences were extracted from the bone marrow (or spleen) filtered_contig.csv files and aligned with IgBlast (*Ye et al., 2013*) before assembling full-length scFvs containing VL and VH separated by a $(Gly_4Ser)_3$ linker and 30 bp homology overhangs on each side, which were ordered from Twist Bioscience. Gibson assembly for each scFv was performed using the NEBuilder HiFi DNA Assembly Master Mix (NEB E2621L) according to the manufacturer's instructions with BamHI linearized pYD1 vector and scFv fragment in 1:4 ratio (in molarity). 5 µl of each reaction including a negative control without scFv was subsequently transformed into 100 µl Fast Transformation of Mix & Go Competent Cells (Zymo T3001) each and plated on LB plates containing Ampicillin. After overnight incubation at 37°C, all plates, except the negative control, had >100 of colonies of which 2 were picked per plate to confirm correct insertion and sequence of scFvs via Sanger sequencing. After confirmation of successful scFv integration into pYD1-BamHI, plasmid DNA was extracted for each sample from a 1 ml overnight culture using the QIAprep Spin Miniprep Kit (QIAGEN 27104). Next, plasmid DNA was transformed into yeast cells using frozen-EZ Yeast Transformation II kit (Zymo T2001). Briefly, 0.2 µg of DNA was mixed with 25 µl of competent cells and 250 µl EZ3 solution. After a 45 min incubation at 37°C, cells were plated onto SD-CAA + 2% glucose plates and incubated at 30°C for 3 days. Per plate, two colonies were inoculated in YNB-CAA + 2% glucose medium overnight at 30°C, followed by 24 hr induction in YNB-CAA + 2% galactose medium prior to flow cytometry screening. Staining of monoclonal scFv yeast cells was performed as described in 'Screening yeast display antibody libraries for antigen binding by FACS'. Briefly, $1 \times 10^6$ cells were stained with 0.02 µg/µl anti-FLAG-PE and 0.01 µg/µl RSV-F-AF647. Yeast cells expressing the RSV-F binding monoclonal antibody palivizumab as scFv were used as positive control (AF647+/PE+) as well as negative control (PE+ only) to set the gates. Flow cytometry scanning to screen for double-positive (AF647+/PE+) cells was performed on BD LSR Fortessa (BD Biosciences).

## Data visualization and statistical analysis

Clonal expansion profiles (*Figure 1d*) displaying the frequencies of clones and clonal fractions were generated using tcR R package (*Nazarov et al., 2015*). Clonal tracking plots (*Figure 5d*, *Figure 5—figure supplement 2*) were generated using R package immunarch (*Nazarov et al., 2021*). Heatmaps (*Figures 1e and 3c*, *Figure 1—figure supplement 2d*, *Figure 3—figure supplement 1b*) were generated using R package pheatmap 1.0.12 (*Kolde, 2019*), all other plots were generated using the R package ggplot2 3.3.2 (*Wickham, 2021*), unless stated otherwise. The following R packages were used for visualization: RColorBrewer R package (*Neuwirth, 2014*) and cowplot (*Wilke, 2020*). GraphPad Prism (version 7) was used for analysis of ELISA data and MAF subsampling analysis (*Figure 1—figure supplement 1a and f*). For comparison between cohorts (*Figure 1b and c*, *Figure 1—figure supplement 1c,d*, *Figure 1—figure supplement 2b*, *Figure 2—figure supplement 2b*), unpaired Student's *t*-test was used, and p-values were corrected for multiple testing using the Benjamini–Hochberg procedure. p-Values <0.05 were considered statistically significant. Violin plots and boxplots represent the median and interquartile range while other statistical data present mean ± SEM. To assess the relationships between clonal frequency and clonal overlap (*Figure 5e*), as well as clonal overlap and antigen specificity (*Figure 5f*), we employed Spearman correlation test (computed with R package ggpubr 0.4.0; *Kassambara, 2020*).

## Acknowledgements

We acknowledge the ETH Zurich D-BSSE Animal Facility, in particular MD Hussherr and Dr. G Camenisch, for excellent assistance with animal experiments and with animal authorization. We acknowledge the ETH Zurich D-BSSE Genomics Facility Basel and Single Cell Unit, in particular E Burcklen, I Nissen, Dr. C Beisel, Dr. M Di Tacchio, and Dr. T Horn, for excellent support and assistance. We thank Mason Minot and Julien Roux for scientific discussions. This work was supported by the European Research Council Starting Grant (Project 679403 to STR) and Swiss National Science Foundation (Project 310030_197941 to STR).

## Additional information

### Competing interests
Simon Friedensohn, Cédric R Weber: Affiliated with Alloy Therapeutics; the author has no financial interests to declare. Sai T Reddy: May hold shares of Alloy Therapeutics and Engimmune Therapeutics; on the scientific advisory board of Alloy Therapeutics and Engimmune Therapeutics. The other authors declare that no competing interests exist.

### Funding

| Funder | Grant reference number | Author |
| --- | --- | --- |
| European Research Council | 679403 | Sai T Reddy |
| Swiss National Science Foundation | 310030_197941 | Sai T Reddy |

The funders had no role in study design, data collection and interpretation, or the decision to submit the work for publication.

### Author contributions
Lucia Csepregi, Conceptualization, Resources, Data curation, Formal analysis, Validation, Visualization, Methodology, Writing – original draft, Writing – review and editing; Kenneth Hoehn, Formal analysis, Visualization, Writing – original draft, Writing – review and editing; Daniel Neumeier, Simon Friedensohn, Methodology; Joseph M Taft, Resources, Methodology; Cédric R Weber, Arkadij Kummer, Formal analysis; Fabian Sesterhenn, Resources; Bruno E Correia, Resources, Supervision; Sai T Reddy, Conceptualization, Resources, Supervision, Funding acquisition, Methodology, Writing – original draft, Project administration, Writing – review and editing

### Author ORCIDs
Lucia Csepregi ⓘ https://orcid.org/0000-0002-0709-4244
Kenneth Hoehn ⓘ https://orcid.org/0000-0003-0411-4307
Joseph M Taft ⓘ https://orcid.org/0000-0003-1345-6122
Bruno E Correia ⓘ https://orcid.org/0000-0002-7377-8636
Sai T Reddy ⓘ https://orcid.org/0000-0002-9177-0857

### Ethics
All mouse experiments were performed under the guidelines and protocols approved by the Basel-Stadt cantonal veterinary office (Basel-Stadt Kantonales Veterinaeramt, Tierversuchsbewilligung #2582).

### Decision letter and Author response
Decision letter https://doi.org/10.7554/eLife.92718.sa1
Author response https://doi.org/10.7554/eLife.92718.sa2

## Additional files

### Supplementary files
• Supplementary file 1. Sequence input into molecular amplification fingerprinting (MAF) pipeline. Numbers of quality-processed and length-trimmed sequencing reads as input into the MAF pipeline for error and bias correction for mice 1x-A, 1x-B, 1x-C, 3x-D, 3x-E, and 3x-F. BM: bone marrow; aLN-L, -R: left and right axillary lymph nodes; iLN-L, -R: left and right inguinal lymph nodes.

• Supplementary file 2. Clonal output after MAF processing. Numbers of unique clones (and clonotypes; defined by antibody sequences possessing identical germline V- and J-genes and 90% CDRH3 a.a. identity and identical length) obtained after MAF processing. BM: bone marrow; aLN-L, -R: left and right axillary lymph nodes; iLN-L, -R: left and right inguinal lymph nodes.

• Supplementary file 3. Overlap of the five most diverse clonotypes across lymphoid organs. Table indicates overlap of the five most diverse clonotypes (top 1–top 5) across lymphoid organs within each mouse. The numbers represent organs (1: aLN-L; 2: iLN-L; 3: iLN-R; 4: aLN-R; 5: spleen; 6: BM)

that share the same clonotypes, being among the top five in all indicated organs. BM: bone marrow; aLN-L, -R: left and right axillary lymph nodes; iLN-L, -R: left and right inguinal lymph nodes.

• Supplementary file 4. B-cell sort and 10× cell yield. Numbers of FACS-isolated B-cell subsets per organ, as well as the yield of all cells and IgG+ B cells after single-cell V(D)J sequencing of antibody repertoires.

• Supplementary file 5. Overlap analysis for the identification of antigen-specific antibody sequences. CDRH3 and CDRL3 information of IgG+ B cells shared among at least three organs, including cell counts for each organ. Black check marks indicate antibodies tested for RSV-F-binding as scFv format in yeast cells. Green check marks indicate confirmed RSV-F-binding of single-cell clones, whereas gray symbols indicate presumptive binding/non-binding of clones belonging to the corresponding clonotype. BM: bone marrow; aLN-L: left axillary lymph node; iLN-L: left inguinal lymph node.

• Supplementary file 6. Primers used for yeast display screening and NGS library preparation. VL-binding primers p3-26 were adapted from *Reddy et al., 2010* and IgG- and VH-binding primers p1, 2, 27–45 were adapted from *Khan et al., 2016*.

• MDAR checklist

## Data availability

High-throughput sequence data is uploaded to the Sequence Read Archive (SRA) with the primary accession code PRJNA763201 (https://www.ncbi.nlm.nih.gov/sra/PRJNA763201). Additional data and code that support the findings of this study are available on the GitHub repository: https://github.com/luciacsep/physiologicalLandscape (copy archived at *Csepregi, 2023*) and https://bitbucket.org/kleinstein/projects.

The following dataset was generated:

| Author(s) | Year | Dataset title | Dataset URL | Database and Identifier |
|---|---|---|---|---|
| Csepregi L, Hoehn KB, Neumeier D, Taft JM, Friedensohn S, Weber CR, Kummer A, Sesterhenn F, Correia BE, Reddy ST | 2021 | IgG VH deep sequencing of *Mus musculus*: lymph nodes, spleen and bone marrow | https://www.ncbi.nlm.nih.gov/sra/PRJNA763201 | NCBI Sequence Read Archive, PRJNA763201 |

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
