## [Editor Report]

This study provides an important systems analysis of antibody repertoires across multiple lymphoid organs, demonstrating significant clonal overlap following repeated immunizations. The findings show that strong humoral responses lead to a high degree of repertoire consolidation, correlating with antigen specificity and B-cell migration between organs. The evidence is convincing, with deep sequencing and network analyses strongly supporting the conclusions.

---

## [Decision Letter]

**Decision letter after peer review:**

Thank you for submitting your article "The physiological landscape and specificity of antibody repertoires is consolidated by multiple immunizations" for consideration by *eLife*. Your article has been reviewed by 3 peer reviewers, and the evaluation has been overseen by a Reviewing Editor and Betty Diamond as the Senior Editor. The following individuals involved in review of your submission have agreed to reveal their identity: Tomoharu Yasuda (Reviewer #3).

Essential revisions (for the authors):

Considering that many biologists are supposed to be interested in this manuscript, authors should try to make the manuscript more easily be understood by the readers. From this points, reviewer request two types of revision. First, bioinformatics analysis should be tried to be more easily understood for readers. Second, authors should discuss the several important points raised by the current analysis.

Reviewers describe requests (in the recommendations for the authors).

In regard to the first point, additional comments of (1), (2) of the reviewer 1, comments (1), (4), (7), (8) of the reviewer 2, and comment (2) of the reviewer 3 should be carefully revised.

In regard to the discussion points, authors should discuss comments (3), (5), (6), and (9) of the reviewer 2, and comments (1) and (4) of the reviewer 3.

*Reviewer #1 (Recommendations for the authors):*

In this study, the authors did in-depth systems analysis of IgG antibody repertoire in immunized mice, and beautifully demonstrated that a limited number of shared clones become dominant in multiple lymphoid organs. However, clonal expansion of antigen-specific memory B cells by multiple immunization, and migration of memory B cells and plasmablasts from one lymphoid tissue to others are known. Extent of clonal expansion of shared clones may depend on immunization protocols that may affect distribution of the antigen to multiple lymphoid organs as the authors discussed. Therefore, the conclusion of this study may not be generalized without testing other immunization protocols.

In this study, the authors showed expansion of overlapping clones across multiple lymphoid organs after repeated immunization. The authors raised distribution of antigens to multiple lymphoid organs and migration of antigen-activated B cells as possible mechanisms for generation of overlapping clones.

It is known that memory B cells are present in circulation as well as various lymphoid organs and home to lymph nodes, and expand by immunization. These previous findings showed migration of memory B cells to different lymphoid organs and their clonal expansion after multiple immunization. Plasmablasts are also known to migrate from spleen and lymph nodes to bone marrow. Therefore, presence of overlapping clones in multiple different lymphoid organs may not be surprising. Although the authors claim a surprisingly high degree of repertoire consolidation, characterized by highly connected and overlapping B-cell clones across multiple lymphoid organs after repeated immunization, extent of clonal expansion may depend on immunization protocols. Overall, it is not so clear what is the novel findings in this study, and whether the findings of this study can be applicable to vaccination in human.

Additional comments

(1) This study generally lacks statistical analyses. Statistical analyses are required when the authors compare two samples and describe that one is lower or higher than the other.

(2) Some figures are similar but different, and how and why they are different are not clearly indicated. Examples are Figure 3c and Supplemental Figure 4b.

*Reviewer #2 (Recommendations for the authors):*

By examining the post-vaccination BCR repertoires in specific lymphoid organs, Csepregi and coworkers provide a fresh look a migration patterns of specific B cell clones. They attempt to convey a vast amount of information through various statistical metrics and data visualization methods developed for repertoire analysis. While this analysis may be a bit heavy and, perhaps even overwhelming, the experimental demonstration that antigen specificity correlates with inter-organ migration convincingly justifies their analysis methods. While the internal consistency of the wet and dry data was impressive, there appears to be room for drawing upon previous work in the literature.

• Summary of what the authors were trying to achieve

This work contains an in-depth analysis of BCR repertoires at the level of multiple lymphoid organs upon vaccination. The results are primarily comprised of bulk sequence analysis, but a single cell sequence dataset is also presented. Moreover, yeast display is used to select antigen binders, and a strong correlation between antigen specificity and inter-organ sharing is observed.

• Major strengths and weaknesses of the methods and results

The main strength was the connection between the first part of the paper (repertoire sharing across organs) and the second part (antigen specificity).

Weaknesses: The bioinformatics analysis is very detailed and took a while to digest. Some of the graphs seemed overly complex for what is actually a pretty intuitive set of results.

• An appraisal of whether the authors achieved their aims, and whether the results support their conclusions

I think that, with the inclusion of the antigen specificity, the authors met or exceeded their aims.

• Likely impact of the work on the field, and the utility of the methods and data to the community

This approach should work for other mouse studies. The antigen specificity aspect may work for human antibody discovery using PBMCs. If sampling is an issue, perhaps pooling of different vaccinated donors' sequence data can be used.

• Any additional context you think would help readers interpret or understand the significance of the work

The data visualization is challenging and probably could use a tutorial. This was my only serious criticism of the work-that I spent a lot of time trying to understand the bioinformatics figures.

I appreciate the novelty and significance of this work, and my impression is overall quite positive. The main area where I see room for improvement is in the descriptions of the bioinformatics analysis. My reading slowed down a lot as I tried to digest some of the figures. It would be nice if the reader could grasp the point of these figures without struggling to understand what is being plotted.

1. Diversity and organ size. You mention that lymph nodes are less diverse but also smaller than the other organs. Since you show, further down in the paragraph, that not only the raw numbers but also the distribution of clone size is skewed toward dominant clones in the LN, it might be helpful to explicitly state that the observed diversity in LN can not explained by their smaller size.

2. Figure 1d. After I worked out what the colors meant, the figure was quite clear. There is a lot going on in the figure: clones are ranked by size, and then binned as noted in the legend, and then the population, as a proportion is plotted. Some explanation to this effect rather than jargon like "repertoire space" or "repertoire polarization" would have helped me a lot.

3. In figure 1e, the difference between 1x (left) and 3x (right) is striking. You refer to this phenomenon as "physiological consolidation". One interpretation is that migration has simply mixed the B cells within the 3x mice creating a more homogeneous repertoire across organs within given mouse than in the 1x cohort. But why do the spleen and BM cluster together in the 1x cohort? Is it correct that this similarity in 1x BM/spleen is not due CDRH3 but to CDR1-2/framework?

4. In figure 2, I found the pie charts to be a little confusing. The colors did not make sense to me. The largest 1x-B aLN-L cluster is mostly orange, which, according to the legend, means "spleen" and "BM". Is the fact that the "aLN-L" color (light blue) does not appear in the pie chart due to there being only 1 or a few aLN-L sequences?

5. Figure 2 is yet another unfamiliar (to me) way of representing data that slowed me down. For example, I could not understand why you say: "in cohort-3x mice clones were frequently shared across lymphoid organs, particularly the lymph nodes (Figure 3a)." Does "frequently shared" refer to the line thickness?

6. You observed "we found that lineage trees of all cohort-3x mice had a significantly higher proportion of transitions both from spleen to bone marrow, and from bone marrow to spleen". It would be helpful from a reader's perspective to put this and other similar observations into context of prior literature. Is this expected? Surprising?

7. In Figure 4, it was hard to distinguish between blue/green colors

8. In Figure 5d, it appeared as if some of the CDRH3 sequences with a given mouse were quite similar. I'm curious as to whether there was convergence across mice. If you compute the clonotype overlap between two mice and then quantified this overlap overall (as a fraction of all clonotypes) and as a fraction of all binders, do you see that the binders overlap more?

9. Could you comment on the extent to which the paired BCR CDRH3 repertoire overlaps with the bulk data? I'm wondering how many of the binders were observed in the 10x dataset.

*Reviewer #3 (Recommendations for the authors):*

Performing the systemic profiling of antibody repertoires across multiple organs within the same individual is important to understand how immune response is regulated to fight against pathogens. In the manuscript entitled "The physiological landscape and specificity of antibody repertoire is consolidated by multiple immunizations", the authors tried to analyze antibody clonality and dynamics upon RS virus antigen immunization to mice by comparing between early time point, 1x immunization, and later time point, 3x immunization. I understood a major finding of this study is a repertoire consolidation across organs after multiple immunizations. Such phenomenon was explained by clonal migration across multiple lymphoid organs, which is directly correlated with antigen specificity. It is interesting and seems important findings contribute to progress in the research field.

The following points should be addressed or improved:

1. p4, Figure 1b, and c; Values of "Unique clones per organs" and "Unique clones per mouse" maybe both mislead readers. Unique clones per organ in Figure 1b is probably not the number per organ. Is not it just counts per input of the arbitrary number of templates from each organ? In addition, how authors were able to calculate unique clones per mouse in 1c? Please clarify the process and reason for calculated values for the accuracy of data.

2. p5, line 144-149; Figure 1d, For the analysis of clonal expansion, authors show clonal frequency rank in which profiles of LN or SP/BM are oppositely changed. However, it makes me hard to interpret data because of the missing unimmunized mouse organ profile. Authors should provide the clonal proportion of unimmunized ones to judge where the baseline is, otherwise, the conclusion here can be speculative.

3. p33 Supplementary Figure 2a; This is not kind for general readers to understand what really figure means. What do Evenness values and α values mean? How should we interpret data to understand the difference between cohort-1x and cohort-3x?

4. Figure 4a and supplementary Figure 2C; Authors described in the method section that Balb/c mice were used in the study. In contrast to C57BL/6 strain, immunoglobulin genes of Balb/c strain have not been well-sequenced, and therefore the database is not completed. Roughly 40% of Ig genomic region is still missing. In IMGT database, many of Balb/c IGHV sequences are not or partially identified. Are authors really sure that SHM was counted on Balb/c reference sequences, not on B6 or other strain? Otherwise, counts and data will be wrong.

[Editors' note: further revisions were suggested prior to acceptance, as described below.]

Thank you for resubmitting your work entitled "The physiological landscape and specificity of antibody repertoires is consolidated by multiple immunizations" for further consideration by *eLife*. Your revised article has been evaluated by Betty Diamond (Senior Editor) and a Reviewing Editor.

The manuscript has been improved but there are some remaining issues that need to be addressed, as outlined below:

*Reviewer #1 (Recommendations for the authors):*

The strength of this study is a systems-based approach that included deep and single-cell sequencing, bioinformatic and statistical analysis, and high-throughput antibody specificity screening to comprehensively profile antibody repertoires from six distinct lymphoid organs. The weakness of this study is that it does not consider the binding properties of major BCR clones, such as affinity for antigen, or the clonal selection of B cells that are broadly reactive to diverse viral variants.

1. The authors properly addressed my concerns by adding untreated mouse profiles, descriptions, figure legend, and updating the analysis based on recently published reference data that strengthened the conclusion of this study and is beneficial for readers.

2. For Figure 1b, "Unique clones per organ" in the y-axis is misleading. I recommend changing the axis such as "Number of unique clones per 135,000 IgG cDNAs".

3. For Figure 1c, "Unique clones per mouse" in the y-axis is misleading. I recommend changing the axis such as "Number of unique clones from 6 organs".

*Reviewer #3 (Recommendations for the authors):*

In this study, the authors did in-depth systems analysis of IgG antibody repertoire in immunized mice, and demonstrated that a limited number of shared clones become dominant in multiple lymphoid organs. Although migration of memory B cells is known, the authors beautifully demonstrate consolidation of the antibody repertoire across different lymphoid organs by multiple immunization. Because the changes in antibody repertoire by immunization may depend on immunization protocols, it is not yet clear whether the findings in this study can be generalized.

In the point-to-point response to the concern of this reviewer on the novelty of the study, the authors listed several findings described in this manuscript as novelty, and mentioned that they edited the manuscript accordingly. However, the introduction and Discussion sections are not much changed. Could the authors make it clear how they edited the manuscript accordingly?

The authors did statistical analyses in some of the data. However, the analyses in Figure 1b, Supplementary Figure 1C, Supplementary Figure 2b and Supplementary Figure 4b are not appropriate. There are more than three data (bars) in each figure, and the authors discuss the comparison of more than three data in the result section. Therefore, the authors should have analyzed the data by multiple comparison test.

---

## [Author Response]

Essential revisions (for the authors):Reviewer #1 (Recommendations for the authors):1) This study generally lacks statistical analyses. Statistical analyses are required when the authors compare two samples and describe that one is lower or higher than the other.

We have now included more statistical analysis for the following figures:

Figure 1b and Supplementary Figure 1c: We performed Welch’s t-test to compare the clonal diversities of organs across cohorts. Only P values of <0.05 are depicted. Analysis showed no statistically significant difference in clonal diversity between lymph nodes of cohort-1x and cohort-3x mice, while it revealed a statistically significant decrease in clonal diversity in spleen of cohort-3x mice (clonal: p=0.016 (Figure 1b), clonotype p=0.0068 (Supplementary Figure 1c)). For bone marrow there was a minor statistically significant increase in regard to clonal (p=0.048) but not clonotype diversity.

Figure 1c and Supplementary Figure 1d: We utilized Welch’s t-test to compare clonal (Figure 1c) and clonotype (Supplementary Figure 1d) diversities within mice across cohorts, depicting only P values <0.05. While clonal diversity showed no significant difference across cohorts, there was a significant difference in clonotype diversity (p=0.012) with cohort-3x mice showing reduced clonotype diversity.

Supplementary Figure 2b: Boxplots indicating the Shannon evenness (α = 1) of each lymphoid organ per cohort (see Methods section p. 19/20 for detailed explanation on evenness analysis). P values represent the results from Welch’s t-test comparing the same organs across both cohorts. Analysis showed all lymph nodes having reduced clonal expansion in cohort-3x mice with statistical significant difference in iLN-L and aLN-R lymph nodes (iLN-L: p=0.037, aLN-R: p=0.019). Both spleen and bone marrow of cohort-3x mice show statistically significant increases in clonal expansion with p values of 0.034 and 1.7e-04, respectively.

Supplementary Figure 4b: We performed Welch’s t-test to compare mean SHM frequencies of organs per cohort. P values represent the results from Welch’s t-test comparing the same organs across both cohorts. Analysis revealed that all organs of cohort-3x mice, except iLN-L (p=0.272), had statistically significant increases in SHM frequencies (aLN-L: p=0.02, iLN-L: p=1.4e-04, aLN-R, spleen: p=1.8e-04, and bone marrow: p=2.6e-03).

Figure 5e: We calculated Spearman correlation to assess the correlation between clonal frequency and clonal organ overlap (N=1-6 organ(s)). We performed this analysis for all clones (indicated in grey), as well as for binding clones only (indicated in green). Analysis revealed correlation between clonal frequency and organ overlap in all three mice (3x-D / 3x-E / 3x-F: R=0.052 / 0.57 / 0.55, p<2.2e-16 / p<2.2e-16 / p<2.2e-16 in all clones and R=0.065 / 0.6 / 0.72, p<2.2e-16 / p<2.2e-16 / p=2.9e-16 in binding clones).

Figure 5f: We computed Spearman correlation to evaluate the correlation between the percentage of detected binding clones and the number of organs they occur (N=1-6 organ(s)). Analysis revealed statistically significant correlation in mice 3x-D and 3x-E with R=1 (p=0.0028) and no statistically significant correlation in mouse 3x-F (R=0.43; p=0.42). A pooled analysis of all three mice showed statistically significant correlation with R=0.79 (p=0.00011), thus revealing a clear positive correlation between organ overlap and detection of antigen-specific clones.

2) Some figures are similar but different, and how and why they are different are not clearly indicated. Examples are Figure 3c and Supplemental Figure 4b.

We appreciate this comment and understand that some analysis / figures require further explanation to clarify their additional value.

Various methods exist for quantifying global antibody repertoire similarities between samples (reviewed in Greiff et al., 2015a; Rosenfeld et al., 2018). For instance, the Jaccard index calculates similarity based on identical clonal overlap across two organ repertoires, while the cosine similarity considers both clonal overlap and clonal frequency (cosine indices near 1 imply high pairwise similarity). Comparing cosine similarities (Figure 3c and Supplementary Figure 6) and Jaccard indices (Figure 3b and Supplementary Figure 5b), we demonstrate consistent patterns of repertoire similarity between two organs, irrespective of the measurement used.

We also restated the corresponding sections to better explain the reasoning of using both approaches, see p.8, first paragraph.

Reviewer #2 (Recommendations for the authors):1. Diversity and organ size. You mention that lymph nodes are less diverse but also smaller than the other organs. Since you show, further down in the paragraph, that not only the raw numbers but also the distribution of clone size is skewed toward dominant clones in the LN, it might be helpful to explicitly state that the observed diversity in LN can not explained by their smaller size.

We agree with this comment and thus modified the corresponding sections (p. 5, second paragraph and p. 6, second paragraph) to clarify that the reduced clonal diversity in lymph nodes cannot be attributed to their smaller size; instead, it likely results from substantial expansion of a few clones following immunization.

In addition, analyzing IgM repertoires of inguinal lymph node, spleen, and bone marrow in three untreated Balb/c mice, we demonstrated comparable clonal diversity of IgM+ repertoires in lymph nodes and bone marrow, with numbers exceeding those observed in the spleen. This emphasizes that the reduced diversity in lymph nodes of cohort-1x and cohort-3x mice is unrelated to their size.

2. Figure 1d. After I worked out what the colors meant, the figure was quite clear. There is a lot going on in the figure: clones are ranked by size, and then binned as noted in the legend, and then the population, as a proportion is plotted. Some explanation to this effect rather than jargon like "repertoire space" or "repertoire polarization" would have helped me a lot.

We restated the corresponding sections for better understanding/clarification (see p. 6, second paragraph and figure legend of Figure 1d).

3. In figure 1e, the difference between 1x (left) and 3x (right) is striking. You refer to this phenomenon as "physiological consolidation". One interpretation is that migration has simply mixed the B cells within the 3x mice creating a more homogeneous repertoire across organs within given mouse than in the 1x cohort. But why do the spleen and BM cluster together in the 1x cohort? Is it correct that this similarity in 1x BM/spleen is not due CDRH3 but to CDR1-2/framework?

We examined the organ repertoire similarities based on V-gene usage (thus including CDR1, CDR2, framework, Figure 1e and Supplementary Figure 2d) as well as CDRH3 identity (see Figure 3c and Supplementary Figure 5b). Both analyses revealed that spleen and bone marrow clustered together, with both organs showing increased diversity based on clonal identity (CDRH3) and unique V-genes compared to lymph nodes in cohort-1x mice. These observations were also confirmed by phylogenetic tree analysis, which additionally indicated a significantly greater proportion of transitions from spleen to bone marrow in cohort-1x mice (Figure 4b and Supplementary Figure 8b).

We believe that the greater proportion of transitions from spleen to bone marrow are the result of B-cell migration from the spleen to the bone marrow, which are physiological connected through the circulatory system. It is well described that plasma cells from secondary lymphoid organs home to the bone marrow which provides a supportive niche allowing long-lived plasma cells to survive and to continuously secrete antibodies (Benner et al., 1981; Slifka et al., 1995). Blink et al. studied the dynamics of an early B-cell response in mice immunized intraperitoneally with protein antigen (NP coupled to KLH) (Blink et al., 2005). They detected NP-specific antibody-secreting cells (ASCs), including high affinity NP-specific ASCs, in the spleen and bone marrow as early as seven days after immunization. Similarly, other studies also showed the presence of NP-specific ASCs in spleen and bone marrow around five and ten days after primary immunization with NP- protein conjugates, respectively (Slocombe et al., 2013; Takahashi et al., 1998). Here, the observed clustering of the spleen and bone marrow in cohort-1x mice support these findings and indicate, at repertoire level, that antigen-specific ASCs may be present in both the spleen and bone marrow ten days after a single immunization. However, it is important to note that repertoire dynamics may vary depending on type of immunogens (e.g., T-cell dependent or independent) and immunization routes. Furthermore, even though these mice were housed under specific-pathogen free conditions, it is possible that previous B-cell responses, e.g., induced by exposure to environmental antigens, could also contribute to the IgG repertoire similarity between the spleen and bone marrow.

Along this line, please also see our response to comment 6.

4. In figure 2, I found the pie charts to be a little confusing. The colors did not make sense to me. The largest 1x-B aLN-L cluster is mostly orange, which, according to the legend, means "spleen" and "BM". Is the fact that the "aLN-L" color (light blue) does not appear in the pie chart due to there being only 1 or a few aLN-L sequences?

Thank you for the comment. We changed the color scheme to avoid overlap with the border color and the pie chart. The border color represents one unique clonotype, meaning all clones (nodes) having the same border color belong to the same clonotype across all organs within one mouse.

The pie charts within the nodes indicate in which organs the specific clone (node) is present (1 color within one circle: this clone is only present in one organ (color code indicates specific organ), two colors (50/50): clone is present in two organs, etc.). All clones (nodes) derived from one organ always display at least one color, namely the color of its corresponding organ within the pie chart; if more colors are visible, that means this exact clone is also shared across other organs. We edited the colors for better discrimination of clonal organ origin.

5. Figure 2 is yet another unfamiliar (to me) way of representing data that slowed me down. For example, I could not understand why you say: "in cohort-3x mice clones were frequently shared across lymphoid organs, particularly the lymph nodes (Figure 3a)." Does "frequently shared" refer to the line thickness?

Thank you for pointing out that this section needs clarification. Both the pie chart as well as the line thickness indicate the degree of clonal sharing / repertoire similarity. While the pie chart depicts the proportion of clones within the corresponding organ that are shared with any other of the six organs within the mouse, the line thickness represents the repertoire similarity between organ pairs (by calculating the Jaccard index between two organs). Both the proportion of shared clones (pie chart) as well as the repertoire similarity (line thickness) are increased in cohort-3x mice, thus indicating increased clonal sharing across organs within these mice.

We restated the corresponding section and added additional explanation for better understanding (p. 8, first paragraph).

6. You observed "we found that lineage trees of all cohort-3x mice had a significantly higher proportion of transitions both from spleen to bone marrow, and from bone marrow to spleen". It would be helpful from a reader's perspective to put this and other similar observations into context of prior literature. Is this expected? Surprising?

Indeed, performing phylogenetic tree analysis (Figure 4, Supplementary Figure 8) revealed a significantly higher proportion of transitions from spleen to bone marrow in cohort-1x mice, whereas cohort-3x mice showed lineage trees with a significantly greater proportion of transitions both from spleen to bone marrow, and from bone marrow to spleen.

The bone marrow has been identified to be the primary site for (long-lived) plasma cells originating from multiple tissues of the body (Manz et al., 2005, 1997; Slifka et al., 1998). Here, cohort-3x mice were immunized three times within 21 day intervals with a highly immunogenic protein antigen. This immunization schedule enabled sufficient time for the generation of memory B cells as well as affinity-matured long-lived plasma cells to be homing to the bone marrow as well as to the spleen and possibly other lymphoid organs (Fooksman et al., 2010).

Additionally, a recent study has challenged the traditional view that all long-lived plasma cells remain resident in the bone marrow (Aaron and Fooksman, 2022; Benet et al., 2021). This study revealed that plasma cells are capable of recirculating between multiple bone marrow niches and even from the bone marrow to the spleen, suggesting that there may be a shared pool of long-lived plasma cells throughout the body (Benet et al., 2021). These findings could offer an additional explanation of the observed transition from spleen to bone marrow as well as from bone marrow spleen and the high degree of repertoire consolidation in cohort-3x mice.

Lastly, in light of prior antibody repertoire studies in humans, the observed connection between spleen and bone marrow is consistent. Meng et al. demonstrated two networks of overlapping B-cell clones – one encompassing blood, bone marrow, lung, and spleen, indicating blood-rich tissue distribution, and another in the gastrointestinal tract (Meng et al., 2017). Similarly, Domínguez Conde et al. found that clones restricted to two tissues, were typically found between the spleen and the liver or bone marrow and enriched in plasma cells, while those spanning over five tissues, including lymph nodes, were abundant in memory B cells (Domínguez Conde et al., 2022). They suggest that tissue-restricted clones may reflect long term immunological memory maintained by long-lived plasma cells in the bone marrow, spleen, and liver.

Collectively, our findings support the spleen / bone marrow axis on a repertoire scale, suggesting B-cell migration through the circulatory and lymphatic systems, antigen drainage, refueling of germinal center responses by repeated antigen exposure, and reactivation of memory B cells (Mesin et al., 2019; Phan and Tangye, 2017) as potential factors contributing to the observed clonal equilibration across all organs, particularly across the spleen and bone marrow.

7. In Figure 4, it was hard to distinguish between blue/green colors

We edited the colors and used different shapes to better distinguish organ origin.

8. In Figure 5d, it appeared as if some of the CDRH3 sequences with a given mouse were quite similar. I'm curious as to whether there was convergence across mice. If you compute the clonotype overlap between two mice and then quantified this overlap overall (as a fraction of all clonotypes) and as a fraction of all binders, do you see that the binders overlap more?

Thank you very much for this interesting comment. We indeed tested for convergence of clonotypes and binding variants (binding clonotypes) across cohort-3x mice. Briefly, we quantified shared clonotypes across all three mice (Author response image 1, left venn diagram), as well as the fraction of shared binding clonotypes (Author response image 1, right venn diagram). Due to the relatively low number of total binding clonotypes (49) it is difficult to make a direct comparison of the overlapping fractions of all clonotypes versus binding clonotypes. However, as convergence is a well reported observation after infection or immunization (Akbar et al., 2021; Ehrhardt et al., 2019; Parameswaran et al., 2013; Trück et al., 2015), we assume if more binders would have been screened, more overlap across binders would have been detected.

**Author response image 1. sa2fig1:** left: Venn diagram depicting numbers of shared clonotypes across cohort 3-x mice. Percentages indicate the ratio of overlapping clones for the corresponding mouse repertoire. Right: Venn diagram depicting numbers of shared binding clonotypes across cohort 3-x mice. Percentages indicate the ratio of overlapping binding clones.

9. Could you comment on the extent to which the paired BCR CDRH3 repertoire overlaps with the bulk data? I'm wondering how many of the binders were observed in the 10x dataset.

Thank you for the comment. We were analyzing the overlap between the bulk (cohort-3x mice) and single-cell data (Sc-3x). Indeed, we found only one overlapping binding clone across bulk cohort-3x and single-cell (Sc-3x) datasets. As previously stated, this is most likely due to the relatively small numbers of B-cell clones sequenced by single-cell sequencing.

**Author response image 2. sa2fig2:** left: Venn diagram depicting numbers of shared CDR3s across cohort-3 mice (n=3, bulk sequenced) and one mouse (Sc-3x; immunized 3x, 10X single-cell sequenced). Right: Venn diagram depicting the numbers of shared binding CDR3s across cohort-3x mice and Sc-3x mouse.

Reviewer #3 (Recommendations for the authors):Performing the systemic profiling of antibody repertoires across multiple organs within the same individual is important to understand how immune response is regulated to fight against pathogens. In the manuscript entitled "The physiological landscape and specificity of antibody repertoire is consolidated by multiple immunizations", the authors tried to analyze antibody clonality and dynamics upon RS virus antigen immunization to mice by comparing between early time point, 1x immunization, and later time point, 3x immunization. I understood a major finding of this study is a repertoire consolidation across organs after multiple immunizations. Such phenomenon was explained by clonal migration across multiple lymphoid organs, which is directly correlated with antigen specificity. It is interesting and seems important findings contribute to progress in the research field.The following points should be addressed or improved:1. p4, Figure 1b, and c; Values of "Unique clones per organs" and "Unique clones per mouse" maybe both mislead readers. Unique clones per organ in Figure 1b is probably not the number per organ. Is not it just counts per input of the arbitrary number of templates from each organ? In addition, how authors were able to calculate unique clones per mouse in 1c? Please clarify the process and reason for calculated values for the accuracy of data.

Thank you for pointing out that this part requires clarification.

In order to ensure accurate and comparable input of material from all 36 organ samples, digital droplet PCR was utilized in the sequencing library preparation protocol to quantify cDNA and intermediate PCR products for each step of the workflow. We used 135.000 cDNA molecules for each sample as input for each sequencing library. Using the same number of molecules within the workflow ensured that we could accurately compare clonal diversity across organs.

After sequencing, all sequences were pre-processed (e.g., removal of low quality sequences, length trimming), error-corrected and annotated. All antibody sequences that possessed the identical V- and J-genes as well as 100% CDRH3 similarity were assigned to one unique B-cell clone. For analysis of clonal diversity per organ, we quantified all unique clones (based on identical V- and J-genes and 100% CDRH3 similarity) present in the corresponding organ repertoire. For analysis of the clonal diversity per mouse, we pooled all clones from the six organs and extracted only the number of unique clones.

For clarification of clonal diversity analysis, we added the clonal definition into the figure legend of Figure 1b.

2. p5, line 144-149; Figure 1d, For the analysis of clonal expansion, authors show clonal frequency rank in which profiles of LN or SP/BM are oppositely changed. However, it makes me hard to interpret data because of the missing unimmunized mouse organ profile. Authors should provide the clonal proportion of unimmunized ones to judge where the baseline is, otherwise, the conclusion here can be speculative.

Thank you for the comment. We added the analysis of IgM repertoires of untreated Balb/c mice, showing that in particular lymph nodes and bone marrow did not exhibit such degrees of clonal expansion [the three most frequent clones occupied 1.2% to 6.8% in lymph nodes and 3.9% to 7.9% in bone marrow of the IgM repertoire]. The spleen of untreated mice exhibited increased clonal expansion compared to lymph nodes and bone marrow with the three most frequent clones showing frequencies of 9.1%, 13.9%, and 11.3% in mouse 1, 2, and 3, respectively. We added this figure in Supplementary Figure 2c and modified the corresponding text section accordingly (p. 6, second paragraph).

3. p33 Supplementary Figure 2a; This is not kind for general readers to understand what really figure means. What do Evenness values and α values mean? How should we interpret data to understand the difference between cohort-1x and cohort-3x?

Thank you for pointing out that this section needs further clarification. We added a detailed explanation of the Hill-based Evenness analysis into the corresponding Methods section (p. 19-20) and restated the main text accordingly, see p. 5, third paragraph. We also added another figure (see Supplementary Figure 2b) to better visualize the differences across cohort-1x and cohort-3x (using Shannon evenness).

We analyzed clonal expansion by using Hill-based Evenness profiles which are adapted from mathematical ecology and allow for comparison of clonal frequency distribution across multiple repertoires (Greiff et al., 2015b; Hill, 1973).

Briefly, Hill-based evenness analysis uses a continuum of diversity indices with the parameter α determining the impact of high frequency clones: with increasing α values, higher frequency clones are weighted more for example, α = 1 represents the Shannon evenness which is commonly used to assess the state of clonal expansion in immune repertoires (Amoriello et al., 2021; Csepregi et al., 2020; Greiff et al., 2017, 2015a; Jost, 2006; Rosenfeld et al., 2018; Venturi et al., 2007). Evenness values range from 0 and 1, with values closer to 1 representing a uniform frequency distribution of clones, while values closer to 0 indicate the prevalence of one or a few highly expanded clones, resulting in a polarized repertoire (see Author response image 3).

Thus, Hill-based evenness profiles allowed us to compare clonal frequency distributions across multiple organ repertoires. It revealed increased clonal expansion in lymph nodes in contrast to spleen and bone marrow in cohort-1x mice, while cohort-3x mice showed high clonal expansion in all organs (Supplementary Figure 2a, b).

**Author response image 3. sa2fig3:** Exemplary Hill-based evenness profiles. Evenness profiles allow visualization of the state of clonal expansion within antibody repertoires. The green line depicts an even (uniform) repertoire in which all clones are equally abundant, while the purple line represents a clonally expanded (polarized) antibody repertoire with one or few highly expanded clones. The dotted line indicates the Shannon evenness (α = 1); with the repertoire indicated in purple showing a low Shannon Evenness ^α=1^E value, resembling a polarized repertoire compared to the green repertoire.

4. Figure 4a and supplementary Figure 2C; Authors described in the method section that Balb/c mice were used in the study. In contrast to C57BL/6 strain, immunoglobulin genes of Balb/c strain have not been well-sequenced, and therefore the database is not completed. Roughly 40% of Ig genomic region is still missing. In IMGT database, many of Balb/c IGHV sequences are not or partially identified. Are authors really sure that SHM was counted on Balb/c reference sequences, not on B6 or other strain? Otherwise, counts and data will be wrong.

We repeated all SHM and phylogenetic tree analyses using a recently published BALB/c IGHV reference database (Jackson et al., 2022). Compared to the previously used IMGT database, we found that using this reference set reduced SHM estimate in ~24.5% of sequences, and increased the SHM estimate in only ~1% of sequences (see Author response image 4). This is consistent with the BALB/c reference set being more accurate than the IMGT reference. Because 1% of sequences with higher SHM under the BALB/c reference may belong to alleles not included in the published BALB/c reference set, we excluded them. We used the remaining sequences to update Figure 4, Supplementary Figure 4 (previous Supplementary Figure 2c), and Supplementary Figure 8 (previous Supplementary Figure 7). Overall, our results did not change with respect to our major conclusions. In particular, our results in Supplemental Figure 8c are clearer, with the left/right (especially the left) lymph node axes standing out more. We have updated the Methods and Results sections to reflect these new changes, and included a link to the source code for reproducing the analyses.

**Author response image 4. sa2fig4:** SHM and junction length estimates using the BALB/c reference versus the IMGT reference databases. Using the BALB/c reference produces overall lower SHM estimates without affecting junction lengths. This shows the BALB/c reference is a better match to the data than the IMGT reference. Sequences above the diagonal line in the middle panel represent 1% of sequences that had higher SHM under the BALB/c database, and were excluded.

References:

Aaron TS, Fooksman DR. 2022. Dynamic organization of the bone marrow plasma cell niche. *The FEBS Journal* 289:4228–4239. doi:10.1111/febs.16385

Akbar R, Robert PA, Pavlović M, Jeliazkov JR, Snapkov I, Slabodkin A, Weber CR, Scheffer L, Miho E, Haff IH, Haug DTT, Lund-Johansen F, Safonova Y, Sandve GK, Greiff V. 2021. A compact vocabulary of paratope-epitope interactions enables predictability of antibody-antigen binding. *Cell Reports* 34. doi:10.1016/j.celrep.2021.108856

Amoriello R, Chernigovskaya M, Greiff V, Carnasciali A, Massacesi L, Barilaro A, Repice AM, Biagioli T, Aldinucci A, Muraro PA, Laplaud DA, Lossius A, Ballerini C. 2021. TCR repertoire diversity in Multiple Sclerosis: High-dimensional bioinformatics analysis of sequences from brain, cerebrospinal fluid and peripheral blood. *EBioMedicine* 103429. doi:10.1016/j.ebiom.2021.103429

Benet Z, Jing Z, Fooksman DR. 2021. Plasma cell dynamics in the bone marrow niche. *Cell Reports* 34:108733. doi:10.1016/j.celrep.2021.108733

Benner R, Hijmans, W, Haaijman JJ. 1981. The bone marrow: the major source of serum immunoglobulins, but still a neglected site of antibody formation. Clin. exp. Immunol. 46, 1-8.

Blink EJ, Light A, Kallies A, Nutt SL, Hodgkin PD, Tarlinton DM. 2005. Early appearance of germinal center–derived memory B cells and plasma cells in blood after primary immunization. *Journal of Experimental Medicine* 201:545–554. doi:10.1084/jem.20042060

Briney BS, Willis JR, Finn JA, McKinney BA, Jr JEC. 2014. Tissue-Specific Expressed Antibody Variable Gene Repertoires. *PLOS ONE* 9:e100839. doi:10.1371/journal.pone.0100839

Csepregi L, Ehling RA, Wagner B, Reddy ST. 2020. Immune Literacy: Reading, Writing, and Editing Adaptive Immunity. *iScience* 23:101519. doi:10.1016/j.isci.2020.101519

Domínguez Conde C, Xu C, Jarvis LB, Rainbow DB, Wells SB, Gomes T, Howlett SK, Suchanek O, Polanski K, King HW, Mamanova L, Huang N, Szabo PA, Richardson L, Bolt L, Fasouli ES, Mahbubani KT, Prete M, Tuck L, Richoz N, Tuong ZK, Campos L, Mousa HS, Needham EJ, Pritchard S, Li T, Elmentaite R, Park J, Rahmani E, Chen D, Menon DK, Bayraktar OA, James LK, Meyer KB, Yosef N, Clatworthy MR, Sims PA, Farber DL, Saeb-Parsy K, Jones JL, Teichmann SA. 2022. Cross-tissue immune cell analysis reveals tissue-specific features in humans. *Science* 376:eabl5197. doi:10.1126/science.abl5197

Ehrhardt SA, Zehner M, Krähling V, Cohen-Dvashi H, Kreer C, Elad N, Gruell H, Ercanoglu MS, Schommers P, Gieselmann L, Eggeling R, Dahlke C, Wolf T, Pfeifer N, Addo MM, Diskin R, Becker S, Klein F. 2019. Polyclonal and convergent antibody response to Ebola virus vaccine rVSV-ZEBOV. *Nature Medicine* 25:1589–1600. doi:10.1038/s41591-019-0602-4

Fooksman DR, Schwickert TA, Victora GD, Dustin ML, Nussenzweig MC, Skokos D. 2010. Development and Migration of Plasma Cells in the Mouse Lymph Node. *Immunity* 33:118–127. doi:10.1016/j.immuni.2010.06.015

Greiff V, Miho E, Menzel U, Reddy ST. 2015a. Bioinformatic and Statistical Analysis of Adaptive Immune Repertoires. *Trends in Immunology* 36:738–749. doi:10.1016/j.it.2015.09.006

Greiff V, Bhat P, Cook SC, Menzel U, Kang W, Reddy ST. 2015b. A bioinformatic framework for immune repertoire diversity profiling enables detection of immunological status. *Genome Med* 7:49. doi:10.1186/s13073-015-0169-8

Greiff V, Menzel U, Miho E, Weber C, Riedel R, Cook S, Valai A, Lopes T, Radbruch A, Winkler TH, Reddy ST. 2017. Systems Analysis Reveals High Genetic and Antigen-Driven Predetermination of Antibody Repertoires throughout B Cell Development. *Cell Reports* 19:1467–1478. doi:10.1016/j.celrep.2017.04.054

Hill MO. 1973. Diversity and Evenness: A Unifying Notation and Its Consequences. *Ecology* 54:427–432. doi:10.2307/1934352

Jackson KJL, Kos JT, Lees W, Gibson WS, Smith ML, Peres A, Yaari G, Corcoran M, Busse CE, Ohlin M, Watson CT, Collins AM. 2022. A BALB/c IGHV Reference Set, Defined by Haplotype Analysis of Long-Read VDJ-C Sequences From F1 (BALB/c x C57BL/6) Mice. *Frontiers in Immunology* 13. Jost L. 2006. Entropy and diversity. *Oikos* 113:363–375. doi:10.1111/j.2006.0030-1299.14714.x

Manz RA, Hauser AE, Hiepe F, Radbruch A. 2005. Maintenance of serum antibody levels. *Annu Rev Immunol* 23:367–386. doi:10.1146/annurev.immunol.23.021704.115723

Manz RA, Thiel A, Radbruch A. 1997. Lifetime of plasma cells in the bone marrow. *Nature* 388:133–134. doi:10.1038/40540

Mathew NR, Jayanthan JK, Smirnov IV, Robinson JL, Axelsson H, Nakka SS, Emmanouilidi A, Czarnewski P, Yewdell WT, Schön K, Lebrero-Fernández C, Bernasconi V, Rodin W, Harandi AM, Lycke N, Borcherding N, Yewdell JW, Greiff V, Bemark M, Angeletti D. 2021. Single-cell BCR and transcriptome analysis after influenza infection reveals spatiotemporal dynamics of antigen-specific B cells. *Cell Reports* 35. doi:10.1016/j.celrep.2021.109286

Meng W, Zhang B, Schwartz GW, Rosenfeld AM, Ren D, Thome JJC, Carpenter DJ, Matsuoka N, Lerner H, Friedman AL, Granot T, Farber DL, Shlomchik MJ, Hershberg U, Prak ETL. 2017. An atlas of B-cell clonal distribution in the human body. *Nat Biotechnol* 35:879–884. doi:10.1038/nbt.3942

Mesin L, Schiepers A, Ersching J, Barbulescu A, Cavazzoni CB, Angelini A, Okada T, Kurosaki T, Victora GD. 2019. Restricted Clonality and Limited Germinal Center Reentry Characterize Memory B Cell Reactivation by Boosting. *Cell* S0092867419313170. doi:10.1016/j.cell.2019.11.032

Parameswaran P, Liu Y, Roskin KM, Jackson KKL, Dixit VP, Lee J-Y, Artiles KL, Zompi S, Vargas MJ, Simen BB, Hanczaruk B, McGowan KR, Tariq MA, Pourmand N, Koller D, Balmaseda A, Boyd SD, Harris E, Fire AZ. 2013. Convergent Antibody Signatures in Human Dengue. *Cell Host and Microbe* 13:691–700. doi:10.1016/j.chom.2013.05.008

Phan TG, Tangye SG. 2017. Memory B cells: total recall. *Current Opinion in Immunology*, Lymphocyte development and activation * Tumour immunology 45:132–140. doi:10.1016/j.coi.2017.03.005

Rosenfeld AM, Meng W, Chen DY, Zhang B, Granot T, Farber DL, Hershberg U, Luning Prak ET. 2018. Computational Evaluation of B-Cell Clone Sizes in Bulk Populations. *Front Immunol* 9. doi:10.3389/fimmu.2018.01472

Slifka MK, Antia R, Whitmire JK, Ahmed R. 1998. Humoral Immunity Due to Long-Lived Plasma Cells. *Immunity* 8:363–372. doi:10.1016/S1074-7613(00)80541-5

Slifka MK, Matloubian M, Ahmed R. 1995. Bone marrow is a major site of long-term antibody production after acute viral infection. *J Virol* 69:1895–1902.

Slocombe T, Brown S, Miles K, Gray M, Barr TA, Gray D. 2013. Plasma Cell Homeostasis: The Effects of Chronic Antigen Stimulation and Inflammation. *The Journal of Immunology* 191:3128–3138.

doi:10.4049/jimmunol.1301163

Takahashi Y, Dutta PR, Cerasoli DM, Kelsoe G. 1998. In Situ Studies of the Primary Immune Response to (4-Hydroxy-3-Nitrophenyl)Acetyl. V. Affinity Maturation Develops in Two Stages of Clonal Selection. *Journal of Experimental Medicine* 187:885–895. doi:10.1084/jem.187.6.885

The Tabula Sapiens Consortium. 2022. The Tabula Sapiens: A multiple-organ, single-cell transcriptomic atlas of humans. *Science* 376:eabl4896. doi:10.1126/science.abl4896

Trück J, Ramasamy MN, Galson JD, Rance R, Parkhill J, Lunter G, Pollard AJ, Kelly DF. 2015. Identification of Antigen-Specific B Cell Receptor Sequences Using Public Repertoire Analysis. *JI* 194:252–261. doi:10.4049/jimmunol.1401405

Venturi V, Kedzierska K, Turner SJ, Doherty PC, Davenport MP. 2007. Methods for comparing the diversity of samples of the T cell receptor repertoire. *Journal of Immunological Methods* 321:182–195. doi:10.1016/j.jim.2007.01.019

Yang F, Nielsen SCA, Hoh RA, Röltgen K, Wirz OF, Haraguchi E, Jean GH, Lee J-Y, Pham TD, Jackson KJL, Roskin KM, Liu Y, Nguyen K, Ohgami RS, Osborne EM, Nadeau KC, Niemann CU, Parsonnet J, Boyd SD. 2021. Shared B cell memory to coronaviruses and other pathogens varies in human age

[Editors’ note: what follows is the authors’ response to the second round of review.]

The manuscript has been improved but there are some remaining issues that need to be addressed, as outlined below:Reviewer #1 (Recommendations for the authors):The strength of this study is a systems-based approach that included deep and single-cell sequencing, bioinformatic and statistical analysis, and high-throughput antibody specificity screening to comprehensively profile antibody repertoires from six distinct lymphoid organs. The weakness of this study is that it does not consider the binding properties of major BCR clones, such as affinity for antigen, or the clonal selection of B cells that are broadly reactive to diverse viral variants.1. The authors properly addressed my concerns by adding untreated mouse profiles, descriptions, figure legend, and updating the analysis based on recently published reference data that strengthened the conclusion of this study and is beneficial for readers.2. For Figure 1b, "Unique clones per organ" in the y-axis is misleading. I recommend changing the axis such as "Number of unique clones per 135,000 IgG cDNAs".3. For Figure 1c, "Unique clones per mouse" in the y-axis is misleading. I recommend changing the axis such as "Number of unique clones from 6 organs".

We have edited the y-axis labels in Figures 1b and c, and Figure 1—figure supplements 1c and d accordingly.

Reviewer #3 (Recommendations for the authors):In this study, the authors did in-depth systems analysis of IgG antibody repertoire in immunized mice, and demonstrated that a limited number of shared clones become dominant in multiple lymphoid organs. Although migration of memory B cells is known, the authors beautifully demonstrate consolidation of the antibody repertoire across different lymphoid organs by multiple immunization. Because the changes in antibody repertoire by immunization may depend on immunization protocols, it is not yet clear whether the findings in this study can be generalized.In the point-to-point response to the concern of this reviewer on the novelty of the study, the authors listed several findings described in this manuscript as novelty, and mentioned that they edited the manuscript accordingly. However, the introduction and Discussion sections are not much changed. Could the authors make it clear how they edited the manuscript accordingly?

We have edited the introduction (last paragraph) and discussion (second and third paragraphs) to reflect the novel findings and to put them into context with previous literature. We marked the changes in yellow.

The authors did statistical analyses in some of the data. However, the analyses in Figure 1b, Supplementary Figure 1C, Supplementary Figure 2b and Supplementary Figure 4b are not appropriate. There are more than three data (bars) in each figure, and the authors discuss the comparison of more than three data in the result section. Therefore, the authors should have analyzed the data by multiple comparison test.

We have reanalyzed the data using an unpaired t-test with multiple testing correction using the Benjamini-Hochberg method. We have updated all relevant plots (Figures 1b and c, Figure 1—figure supplements 1c and d, Figure 1—figure supplement 2b, and Figure 2—figure supplement 2b) and figure legends accordingly.